# Iron and manganese co-limit the growth of two phytoplankton groups dominant at two locations of the Drake Passage

Jenna Balaguer [1,2✉], Florian Koch[2,3], Christel Hassler[4,5] & Scarlett Trimborn [1,2]

While it has been recently demonstrated that both iron (Fe) and manganese (Mn) control Southern Ocean (SO) plankton biomass, how in particular Mn governs phytoplankton species composition remains yet unclear. This study, for the first time, highlights the importance of Mn next to Fe for growth of two key SO phytoplankton groups at two locations in the Drake Passage (West and East). Even though the bulk parameter chlorophyll *a* indicated Fe availability as main driver of both phytoplankton assemblages, the flow cytometric and microscopic analysis revealed FeMn co-limitation of a key phytoplankton group at each location: at West the dominant diatom *Fragilariopsis* and one subgroup of picoeukaryotes, which numerically dominated the East community. Hence, the limitation by both Fe and Mn and their divergent requirements among phytoplankton species and groups can be a key factor for shaping SO phytoplankton community structure.

[1] Marine Botany, University of Bremen, 28359 Bremen, Germany. [2] Ecological Chemistry, Alfred Wagener Institute Helmholtz Centre for Polar and Marine Research, 25570 Bremerhaven, Germany. [3] University of Applied Science Bremerhaven, 27568 Bremerhaven, Germany. [4] Swiss Polar Institute, Ecole Polytechnique Fédérale de Lausanne, Lausanne, Switzerland. [5] Department F.-A. Forel for Environmental and Aquatic Sciences, University of Geneva, Geneva, Switzerland. ✉email: jenna.balaguer@awi.de

The Southern Ocean (SO) is unique compared to much of the global ocean as it accounts for over 40% uptake of anthropogenically derived carbon dioxide ($CO_2$), making it the largest oceanic $CO_2$ sink[1,2]. With 10% being biologically fixed[3], the amount of biological sequestered $CO_2$ by SO phytoplankton is still underestimated[4]. Even with perennial high macronutrients concentrations ($NO_3$, $PO_4$, $SiOH_4$), primary production remains low in the SO[5], making the SO the world's largest High-Nutrient Low-Chlorophyll (HNLC) region. In addition to macronutrients, trace metals are also needed for several physiological processes and therefore influence phytoplankton growth[6]. It has been demonstrated that in particular the scarcity of the trace metal iron (Fe) drives Antarctic phytoplankton growth and community composition[7–9] but was however measured in low concentrations in SO waters[10–12]. Phytoplankton cells have a high Fe requirement as Fe is involved in various cellular processes such as carbon and nitrogen fixation, nitrate and nitrite reduction, chlorophyll synthesis and especially photosynthesis and respiration processes[6,13]. Some studies have shown that under Fe limitation the transfer efficiency of excitation energy in photosystem II (PSII) decreases[14]. Next to Fe, manganese (Mn) is the second most abundant trace metal needed during photosynthesis[15] where it serves as a co-factor in the oxygen-evolving complex and enables the transport of electrons to photosystem II[16]. It is also vital for the Mn-containing antioxidant enzyme superoxide dismutase, which scavenges reactive oxygen species during photosynthesis, thus preventing potential cell damage[17]. Similar to Fe, total dissolved Mn (dMn) concentrations were found to be very low in the Atlantic sector of the SO (0.04 nM[18]). Also, low Mn concentrations have been reported in several regions of the SO such as the Scotia Sea, Weddell Sea, Ross Sea and the Drake Passage[11,18–23]. Several trace metal (TM) bottle enrichment experiments in the SO have indicated the occurrence of a Mn co-limitation with Fe of phytoplankton[19,21–23]. Across the Ross Sea, various natural *Phaeocystis* populations were found to produce protein signatures indicating late-season Mn and Fe stress, a response that was accompanied by significant chlorophyll build-up upon Mn or Fe additions[22]. Growth and species composition of natural phytoplankton assemblages from the Weddell-Scotia Confluence were influenced by enrichment with Fe or Mn alone[19] while the addition of both TMs, however, was not tested. Various natural phytoplankton assemblages across the Drake Passage increased their photosynthetic activity and biomass after Fe, Mn and ash additions (containing both Fe and Mn) compared to additions of only Fe[21]. Together with the measurements of very low Mn concentrations at these sampling locations, Browning et al.[21] speculated that these phytoplankton communities were co-limited by Mn and Fe. Only recently, clear evidence for Fe-Mn co-limitation of phytoplankton growth in the Drake Passage was provided by Browning et al.[23], a hypothesis previously proposed by Martin et al.[7]. Due to different TM availability in the seawater and different TMs requirements by the cells, phytoplankton can be limited by multiple TMs[6,24] this leading to co-limitation by different TMs at the community level[25]. A strict definition of nutrient co-limitation was given by Liebig's law[26] describing that an organism is co-limited when one nutrient is limiting followed by a second nutrient limitation. This definition was, however, refined by Saito et al.[27], differentiating between three types of co-limitation: Type I: independent nutrient co-limitation; Type II: biochemical substitution co-limitation and Type III: biochemical dependent co-limitation. However, no concomitant effect of Fe and Mn limitation on SO phytoplankton species composition was found in the study by Browning et al.[23], potentially due to the short incubation time between 2 up to maximal 5 days. Hence, clear evidence of how Mn availability influences SO phytoplankton growth at the species level and thereby shapes SO

phytoplankton composition is still lacking in the field. A laboratory study using the bloom-forming SO diatom *Chaetoceros debilis* showed that only the supply of Fe and Mn together led to optimal growth, photochemical efficiency and carbon production[28].

To unravel how Mn availability shapes SO phytoplankton distribution we investigated, the potential limiting or co-limiting effects of both Fe and Mn on growth and species composition of SO phytoplankton in the field. Our results show that in two natural communities of the Drake Passage the growth of two key members of the phytoplankton community (*Fragilariopsis* sp. and picoeukaryotes) was limited by both Fe and Mn while the other phytoplankton groups were primarily limited by Fe.

## Results

### Chemical and biological characteristics of the two sampling stations West and East

Both sampling locations West and East (Fig. 1), where Fe-Mn addition experiments were performed, were typical SO HNLC stations, characterized by high concentrations of macronutrients ($NO_x$ = nitrate ($NO_3$) + nitrite ($NO_2$); phosphate ($PO_4$); silicate ($SiOH_4$)) and low concentrations of chlorophyll *a* (Chl*a*), dissolved Fe (dFe), dissolved Mn (dMn), dissolved zinc (dZn) and dissolved cobalt (dCo) (Table 1). At the sampling location West, among the microphytoplankton (Fig. 2a; Supplementary Table 1) the diatom *Fragilariopsis* sp. was most abundant (306 cells mL$^{-1}$, 95% of total cells), whereas the diatom genera *Chaetoceros* sp. (6 cells mL$^{-1}$, 2%), *Pseudo-nitzschia* sp. (3 cells mL$^{-1}$, 1%) as well as the haptophyte *Phaeocystis antarctica* (6 cells mL$^{-1}$, 2%) were only found in low cell abundances. Hence, at location West the Shannon diversity index of the microphytoplankton group was low with 0.24. At the sampling location East, among the microphytoplankton (Fig. 2c; Supplementary Table 1) *Fragilariopsis* sp. was also the most abundant diatom (230 cells mL$^{-1}$, 77%), followed by *Chaetoceros* sp. (43 cells mL$^{-1}$, 14%), *Pseudo-nitzschia* sp. (20 cells mL$^{-1}$, 7%) and *P. antarctica* (7 cells mL$^{-1}$, 2%). Here, the total Shannon diversity index of the microphytoplankton was high, being 0.75. For both experiments, picoeukaryotes (P), nanoeukaryotes (N) and heterotrophic bacteria (B) were determined via flow cytometry, with the three subgroups of P defined according to their sizes (P1, P2 and P3) (Supplementary Fig. 1). At both locations, P were present (Fig. 2b, d; Supplementary Table 2). Among P, the P2 group (1332 and 701 cells mL$^{-1}$) reached highest abundances relative to the total P abundance at the station West and East, respectively. N as well as B were also found at both locations with almost similar abundances of both groups for the location West and East (Table 2).

### Development of the community composition over both experiments

During both experiments, macronutrients were not limiting, remaining in excess from the start until the end, with concentrations at the end of the experiment being higher than 22 µmol L$^{-1}$ for NOx, 1.3 µmol L$^{-1}$ for $PO_4$ and 14 µmol L$^{-1}$ for $SiOH_4$ (Supplementary Table 3). Chl*a*-based accumulation rates, growth rates of three P groups derived from flow cytometry as well as of the different microphytoplankton genera based on light microscopy were calculated using Eq. (1) (cf. 'Methods' section).

*Experiment West*. For all (total) and the large cells (>2 µm) (Fig. 3a), the addition of Fe or together with Mn led to a significant increase of the Chl*a*-based accumulation rates compared to the Control and the +Mn treatment. For the small cells (<2 µm), Chl*a*-based accumulation rates of the Control were higher relative to the +Mn, lower relative to the +Fe and similar to the +FeMn treatment. In comparison, the Chl*a*-based accumulation rates of the +FeMn were lower than for the +Fe treatment. Three

P subgroups were determined via flow cytometry (Fig. 3b). Among them, the net growth rate of all three subgroups (P1–3) remained unaltered after addition of Mn alone relative to the Control treatment. Only when Fe or FeMn together were added, the growth rate of all P groups was significantly increased relative to the Control. Among the microphytoplankton (Fig. 3c), the net growth rate of *Chaetoceros* sp. did not change after addition of Mn, Fe or both TMs together compared to the Control. The growth rate of *Fragilariopsis* sp. was similar between the Control and the +Mn treatment, but was significantly increased after addition of +Fe or +FeMn. In comparison, the addition of FeMn together resulted in a significant increase of growth of *Fragilariopsis* sp. from 0.08 to 0.11 day⁻¹ relative to the +Fe addition. Growth rates of *Pseudo-nitzschia* sp. and *P. antarctica* were similar between the Control and the +Mn treatment. In contrast, the +Fe and +FeMn additions led to a significant increase of the growth rate of *Pseudo-nitzschia* sp. relative to the Control. Similarly, relative to the Control, a significant increase in growth was observed for *P. antarctica* after the single addition of Fe, but not after FeMn addition. The abundances of heterotrophic bacteria (B) remained unchanged between all treatments (Table 2). N abundances were similar across the different treatments, being generally low (Table 2).

*Experiment East.* The Chl*a*-based accumulation rates of the total and the large cells (> 2 μm) were similar between the Control and +Mn treatment (Fig. 4a). Only for the small fraction (0.2–2 μm), the accumulation rate of the +Mn treatment was significantly

reduced relative to the Control. Relative to the Control, Chl*a*-based accumulation rates (Fig. 4a) of all size fractions (total, small and large) were significantly enhanced after addition of Fe and of FeMn. When Fe and Mn were added together, the Chl*a*-based accumulation rate of all and small cells significantly increased from 0.02 to 0.14 day⁻¹ relative to the addition of Fe alone. For the P1 group (Fig. 4b), each TM addition significantly decreased growth relative to the Control. In contrast, the addition of Mn alone did not change the growth rate of P2 relative to the Control. After addition of both trace metals together and of Fe alone, growth of the P2 group was significantly increased relative to the control. The addition of TMs together led to a significant growth increase of P2 from 0.11 to 0.13 day⁻¹ compared to the single addition of Fe. With respect to P3, the addition of Fe and of both TMs resulted in higher growth rates compared to the Control and the +Mn treatments. For the microphytoplankton (Fig. 4c), the growth rate of *Fragilariopsis* sp. was similar between the Control and +Mn treatments, while +Fe and +FeMn resulted in significant increases relative to the Control. The growth rate of *Chaetoceros* sp. decreased after the addition of Mn, remained unchanged after +Fe addition, but significantly increased after addition of +FeMn compared to the Control. Adversely, compared to the Control the +Fe and +FeMn additions led to a significant increase of growth rates of *Pseudo-nitzschia* sp. and *P. antarctica*. For both species, no changes in growth were found after single addition of Mn. At the end of the experiment, the abundance of B did not change across treatments (Table 2). For N, also no changes were observed between the four different treatments (Table 2).

**Photophysiological responses**
*Experiments.* At the end of both experiments, the maximum quantum yield of PSII (Fᵥ/Fₘ) of the Control and the +Mn treatment were similar (Table 3). For both experiments, the addition of Fe alone triggered a significant increase relative to the Control while only the addition of both trace metals together yielded maximum Fᵥ/Fₘ values (West: 0.51 ± 0.04; East: 0.47 ± 0.02, Table 3; Supplementary Fig. 3) significantly higher than the single addition of Fe. The functional absorption cross-sections of PSII ($\sigma_{PSII}$) remained generally unchanged in response to the different experimental conditions (Table 3). At the end of experiment West, neither the addition of +Mn nor of +Fe altered $\sigma_{PSII}$ relative to the Control, with ranging between 4.3 and 4.5 nm² (Table 3). Only when Fe and Mn were added together, miminum $\sigma_{PSII}$ values were recorded, being 2.80 ± 1.24 nm²

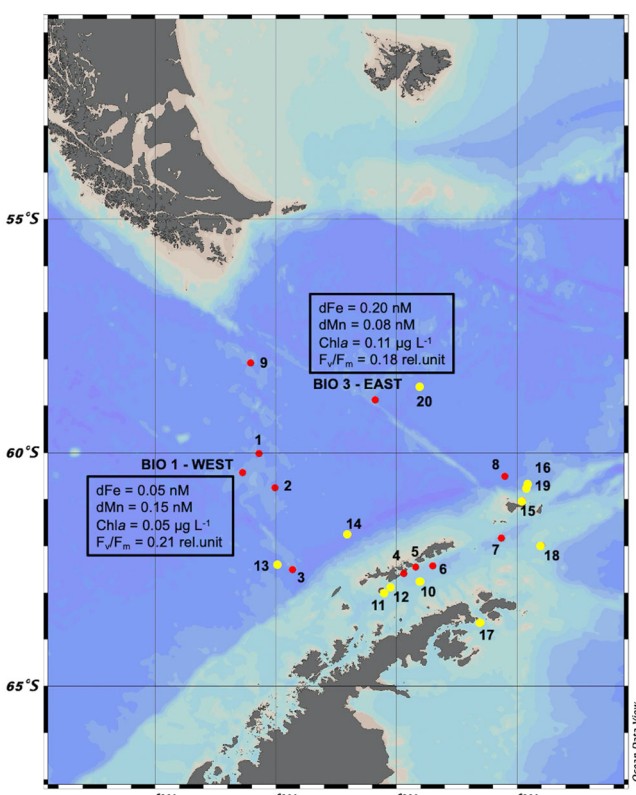

**Fig. 1 Overview of the sampling area in the Drake Passage and the Western Antarctic Peninsula.** Twenty-two in situ stations were sampled during the Polarstern expeditions PS97 (red dots), including BIO1 as station West and BIO3 as station East, as well as during PS112 (yellow dots). For station West and East, initial concentrations of dissolved iron (dFe), dissolved manganese (dMn), chlorophyll *a* (Chl*a*) as well as the 1h-dark-adapted maximum photosystem II quantum yield (Fᵥ/Fₘ) are shown in the dark box.

| Table 1 Initial conditions. | | | |
|---|---|---|---|
| | | **West** | **East** |
| Latitude | min⁻¹ | 60° 24.78′ S | 58° 52.17′ S |
| Longitude | min⁻¹ | 66° 21.85′ W | 60° 51.92′ W |
| dFe | nM | 0.05 | 0.20 |
| dMn | nM | 0.15 | 0.08 |
| dZn | nM | 3.21 | 1.71 |
| dCo | nM | 0.03 | 0.02 |
| dFe:dMn | nM: nM | 0.33 | 2.50 |
| dFe:dZn | nM: nM | 0.02 | 0.12 |
| dFe:dCo | nM: nM | 1.67 | 2.50 |
| NOₓ | μM | 24 | 27 |
| PO₄ | μM | 1.5 | 1.8 |
| SiOH₄ | μM | 17 | 22 |
| Chl*a* | μg L⁻¹ | 0.05 | 0.10 |

Concentrations of total dissolved iron (dFe), dissolved manganese (dMn), dissolved zinc (dZn), dissolved cobalt (dCo), macronutrients (NOₓ = NO₃ (nitrate) + NO₂ (nitrite), PO₄ = phosphate, SiOH₄ = silicate) and chlorophyll *a* (Chl*a*) were determined in the seawater sampled West and East of the Drake Passage.

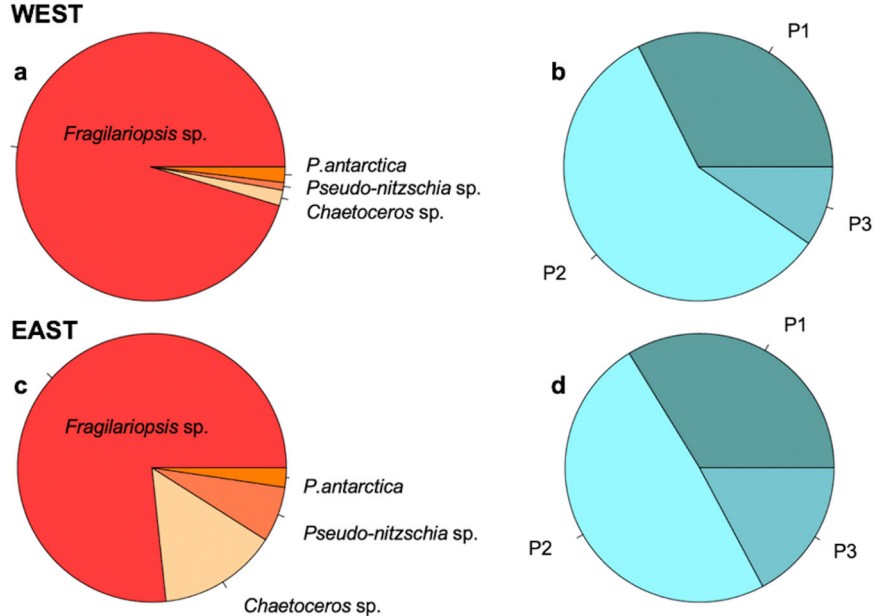

**Fig. 2 Initial relative abundances of the different phytoplankton groups.** Relative abundances of the four microphytoplankton genera (**a**, **c**; based on data of Supplementary Table 1) and of the three picophytoplankton subgroups (P1–P3; **b**, **d**; based on data of Supplementary Table 2) at the start of the experiment at Station West (**a**, **b**) and East (**c**, **d**) was determined via light microscopy (**a**, **c**) and flow cytometry (**b**, **d**).

(Table 3). For the experiment East, $\sigma_{PSII}$ of the final communities of the Control was significantly increased from ~3.4 to 4.7 nm$^2$ (Table 3), but only after addition of Mn alone. The time constant for electron transfer at PSII ($\tau_{QA}$) of the final communities was in most cases not altered in response to the different experimental conditions (Table 3). Only after addition of both trace metal together, $\tau_{QA}$ was significantly enhanced relative to the Control, but this effect was only found for the experiment West. The connectivity between adjacent photosystems (P) of the final communities from both experiments did not change between the Control and +Mn treatment (Table 3). In comparison, P was significantly increased following the addition of Fe alone (Table 3) and slightly after the addition of both trace metals together compared to the Control (Table 3).

*In situ stations.* In addition to the two sampling experiment locations West and East, where the two Fe-Mn addition experiments were performed, 9 stations (Fig. 1) were sampled during the same cruise (PS97) to determine in situ dFe and dMn concentrations in surface Drake Passage waters. In order to complement our results, 11 additional sampling stations across the Drake Passage from PS112 (Fig. 1) were added, for which $F_v/F_m$ as well as concentrations of dFe and dMn were determined (Table 4). Across the 22 sampling sites (this cruise and PS112), a wide range of dMn and dFe concentrations (0.05–~7 nM) as well as of $F_v/F_m$ values (0.12–0.56) was found (Tables 3, 4). Tests for linear regression and correlation were performed between dTMs and $F_v/F_m$ for the 22 stations (Fig. 5, Table 4), revealing that $F_v/F_m$ was not affected by dFe (Fig. 5a; $R^2 \sim 0$, $p > 0.05$) and therefore there was no correlation between dFe and $F_v/F_m$ (Fig. 5a; $r \sim 0.2$, $p > 0.05$). However, 50% of the $F_v/F_m$ response can be explained by dMn concentrations (Fig. 5b; $R^2 \sim 0.5$, $p < 0.005$) and a positive and significant moderate correlation between dMn and $F_v/F_m$ was found (Fig. 5b; $r = 0.7$, $p < 0.005$).

## Discussion
This study shows evidence that next to Fe, Mn availability is an additional driver of SO phytoplankton composition, acting as

**Table 2 Cell abundances of heterotrophic bacteria and nanoeukaryotes.**

|  | Heterotrophic bacteria cells $10^5$ mL$^{-1}$ | Nanoeukaryotes cells mL$^{-1}$ |
|---|---|---|
| *West* |  |  |
| Initial | 3.0 | 37.2 |
| Control | 4.0 ± 0.7$^a$ | 8.3 ± 7.6$^a$ |
| +Mn | 3.9 ± 0.1$^a$ | 25.5 ± 31.5$^a$ |
| +Fe | 4.3 ± 0.8$^a$ | 93.0 ± 70.3$^a$ |
| +FeMn | 4.1 ± 0.2$^a$ | 74.4 ± 56.6$^a$ |
| *East* |  |  |
| Initial | 3.0 | 87.2 |
| Control | 3.9 ± 0.2$^a$ | 243.6 ± 102.2$^a$ |
| +Mn | 2.9 ± 0.7$^a$ | 193.7 ± 45.2$^a$ |
| +Fe | 3.8 ± 0.8$^a$ | 345.1 ± 51.1$^a$ |
| +FeMn | 4.5 ± 0.4$^a$ | 370.0 ± 76.4$^a$ |

Both were determined via flow cytometry at the beginning and the end of the experiment from the communities sampled West and East after exposure to different Fe and Mn availabilities. Values represent the mean ± SD ($n = 6$). Different letters indicate significant differences between treatments ($p < 0.05$).

promoter for growth of specific groups within the same phytoplankton community. For two sites of the Drake Passage, we can show that growth of the diatom *Fragilariopsis* sp., being the most abundant species among the microphytoplankton community at the location West, as well as picoeukaryotes, which dominated the whole community at the location East, were Fe-Mn co-limited while the other phytoplankton groups were primarily limited by Fe.

Both experiments were conducted with typical SO HNLC seawater[29,30], characterized by high macronutrient concentrations (Table 1) while Chl*a* concentrations (Table 1) and the photophysiological status $F_v/F_m$ were low (Table 3). Concentrations of dMn at our two sampling sites (Table 1) were low, being below 0.20 nM and similar to previously measured values across the Drake Passage[18,20,21,23]. At both sites, the dFe concentrations were also low (West: 0.05 nM, East: 0.20 nM, Table 1) and indicative of Fe-limited HNLC waters (~0.20 nM)[31]. Accordingly, cell abundances were low, with the initial phytoplankton communities being

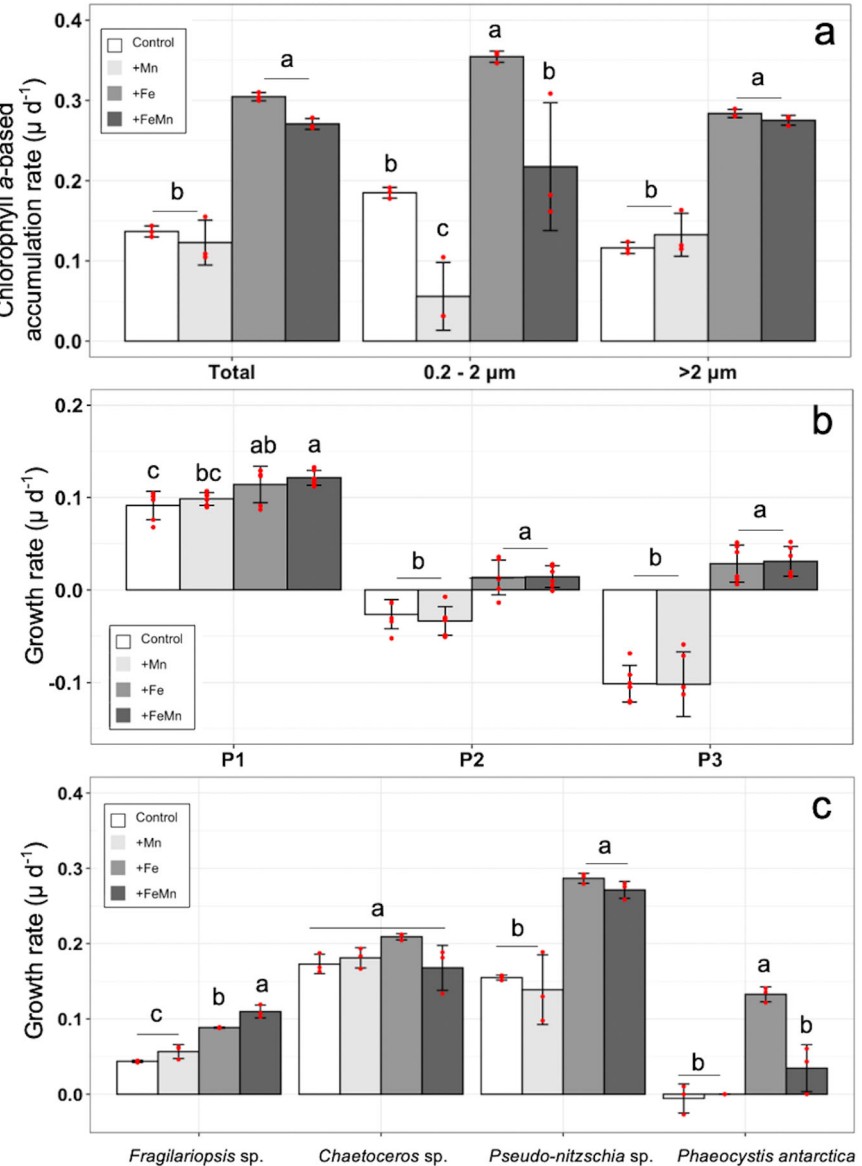

**Fig. 3 Community response at Station West. a** Chlorophyll *a*-based accumulation rate (d$^{-1}$) of all (total), the small (0.2–2 µm) and the large (>2 µm) cells (*n* = 3), **b** net growth rate of each picoeukaryote group (subgroups P1–P3) determined via flow cytometry (*n* = 6) and **c** net growth rate of *Chaetoceros* sp., *Fragilariopsis* sp., *Pseudo-nitzschia* sp. and *Phaeocystis antarctica* determined via light microscopy (*n* = 3). All parameters were estimated from the community sampled at station West (BIO1) after exposure to different Fe and Mn availabilities (Supplementary Fig. 2). Values represent the mean ± SD. Different letters indicate significant differences between treatments (*p* < 0.05).

numerically dominated by three P subgroups (West: ~2300 cells mL$^{-1}$, East: ~1500 cells mL$^{-1}$, Fig. 2, Supplementary Table 2). As typically observed for open SO sites[30,32,33], among the microphytoplankton (> 2 µm), diatoms were found to be the dominant group at both locations (Fig. 2, Supplementary Table 1). The heavy silicified *Fragilariopsis* sp. was numerically the most abundant species among diatoms at both sites (Fig. 2, Supplementary Table 1), as previously reported in SO HNLC waters[34]. Biomass, plankton abundances and dTM concentrations were all low, suggesting that next to Fe, Mn could also play a role in shaping SO phytoplankton composition.

After 2 weeks of incubation, final cell numbers of the heterotrophic bacteria remained constant across all treatments in both experiments (~4–5 × 10$^5$ cells mL$^{-1}$, Table 2). Hence, this group was not influenced by any of our trace metal additions, pointing towards a high ability of this group to cope with changes in trace metal supply. Indeed, during the same expedition, Blanco-

Ameijeiras et al.[35] performed Fe-ligand addition experiments with the same two natural plankton assemblages at station West and East. Their study showed that exopolymeric substances additions stimulated growth of heterotrophic bacteria at location West and East, most likely alleviating dissolved organic carbon limitation. Similarly, other studies have also shown that while low Fe concentration had no impact on heterotrophic bacterial communities[36,37], carbon, and not TMs, may limit their growth in HNLC regions[38,39]. In contrast to the bacteria, distinct effects in phytoplankton species composition were found in response to the different Fe and Mn availabilities in both experiments. Similar to other experiments in HNLC waters[8,40–42], the Fe addition significantly enhanced Chl*a* build-up in both experiments (Figs. 3, 4). As up to 23–24 atoms of Fe are needed in both photosystems (PSI and PSII) for a single copy of the electron transport chain, 80% of the cellular Fe is required for photosynthetic electron transport[13,15]. Hence, Fe addition can increase

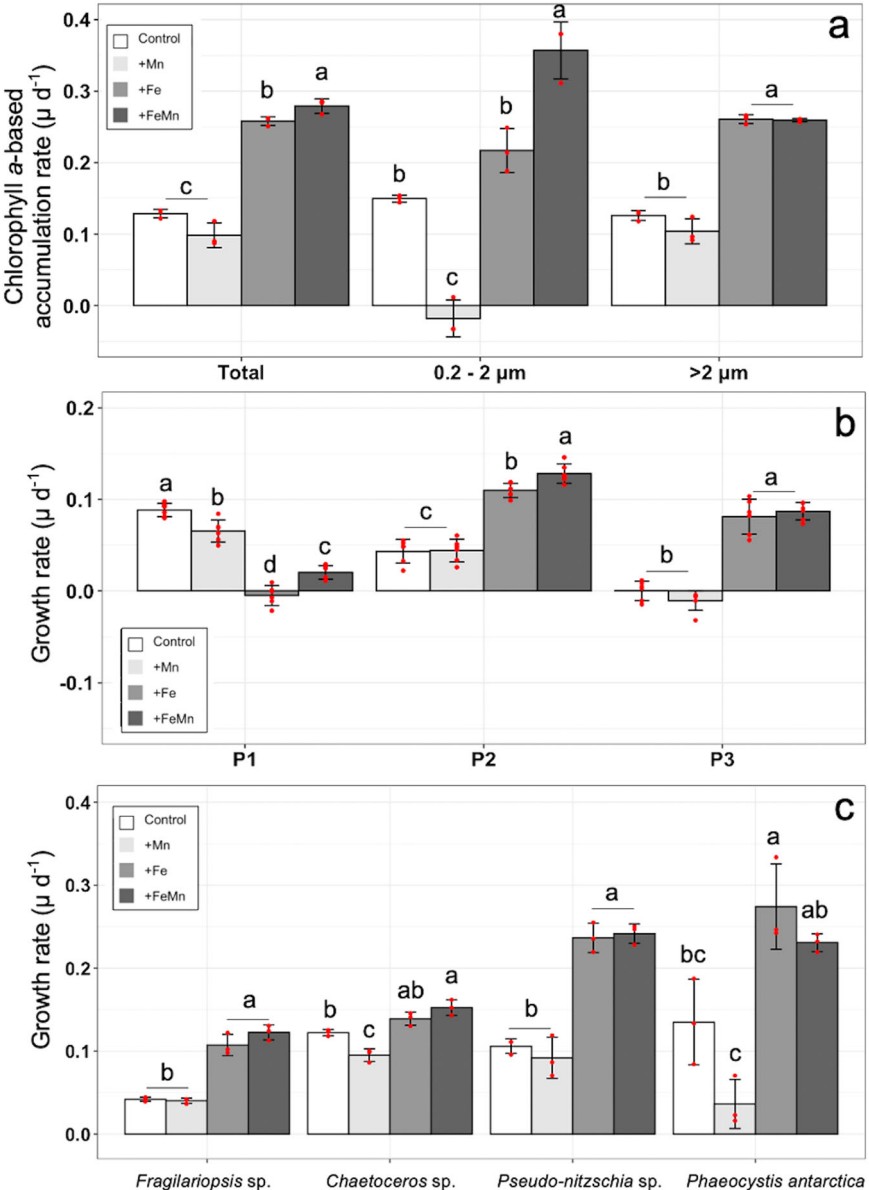

**Fig. 4 Community response at Station East. a** Chlorophyll *a*-based accumulation rates (d$^{-1}$) of all (total), the small (0.2–2 µm) and the large (>2 µm) cells ($n = 3$), **b** net growth rate of each picoeukaryote group (three subgroups P1–P3) determined via flow cytometry ($n = 6$) and **c** net growth rate of *Chaetoceros* sp., *Fragilariopsis* sp., *Pseudo-nitzschia* sp. and *Phaeocystis antarctica* determined via light microscopy ($n = 3$). All parameters were estimated at the end of the experiment from the community sampled at station East (BIO3) after exposure to different Fe and Mn availabilities (Supplementary Fig. 2). Values represent the mean ± SD. Different letters indicate significant differences between treatments ($p < 0.05$).

the Chl*a* content, reliefing chlorosis[43–46]. Indeed, Chl*a* accumulation was promoted at both locations after addition of Fe alone and as a consequence growth of P and microphytoplankton was promoted, except for P1 at location East (Figs. 3a, 4a), indicating that Fe was the main driver at the community level. Even though for experiment West, the analysis of bulk parameters such as Chl*a* indicated Fe availability as main driver, the microscopic analysis, however, revealed that the ecologically and biogeochemically important diatom *Fragilariopsis* sp., which dominated the microphytoplankton community, significantly increased its growth by 0.03 day$^{-1}$ only when Fe was added in conjunction with Mn compared to the single addition of Fe (Fig. 3c). In fact, the cell abundance of *Fragilariopsis* sp. was enhanced by ~35% in the +FeMn relative to the +Fe treatment (Supplementary Table 1). This suggests a relief of FeMn co-limitation of the dominant diatom *Fragilariopsis* sp. at West. However, no FeMn

co-limitation on growth of *Fragilariopsis* sp. was observed at location East. It is possible that *Fragilariopsis* was either biochemically co-limited by Fe and another nutrient (Type III[27]) or was not co-limited at this location. As location West displayed 4 times lower dFe concentration than location East, the requirements of *Fragilariopsis* for other TMs like Mn could have been increased. An increased requirement for other TMs (Mn, Zn, Co) was already reported under Fe limitation for the SO diatom *Chaetoceros simplex*, the Antarctic cryptophyte *Geminigera cryophila*[47] and the two temperate diatoms *T. pseudonana* and *T. oceanica*[48]. In addition, due to the reason that *F. curta*, *F. nana*, *F. cylindrus* and *F. pseudonana* are very difficult to distinguish under the light microscope[49], we grouped these species together as *Fragilariopsis* sp.. Therefore, the ratio between them at both locations is unknown. We speculate that the TMs requirements by the same or different *Fragilariopsis* species may potentially

**Table 3 Photophysiological responses.**

|  | $F_v/F_m$ | $\sigma_{PSII}$ | $\tau_{QA}$ | $P$ |
|---|---|---|---|---|
|  | rel. unit | $nm^2$ | $\mu s$ | rel. unit |
| *West* |  |  |  |  |
| Initial | 0.21 ± 0.02 | 5.61 | 588 | 0.25 |
| Control | 0.26 ± 0.04[c] | 4.46 ± 1.00[a] | 470 ± 58[a] | 0.13 ± 0.05[c] |
| +Mn | 0.25 ± 0.04[c] | 4.20 ± 0.73[a] | 536 ± 57[a] | 0.18 ± 0.06[c] |
| +Fe | 0.44 ± 0.05[b] | 4.26 ± 1.51[a] | 643 ± 11[a] | 0.36 ± 0.06[b] |
| +FeMn | 0.51 ± 0.04[a] | 2.80 ± 1.24[b] | 721 ± 60[b] | 0.35 ± 0.03[a] |
| *East* |  |  |  |  |
| Initial | 0.18 ± 0.03 | 4.68 | 290 | 0.10 |
| Control | 0.25 ± 0.07[c] | 3.42 ± 1.15[b] | 669 ± 280[b] | 0.21 ± 0.01[c] |
| +Mn | 0.27 ± 0.01[c] | 4.68 ± 0.52[a] | 598 ± 46[a] | 0.24 ± 0.02[c] |
| +Fe | 0.43 ± 0.05[b] | 3.22 ± 1.52[b] | 711 ± 66[b] | 0.36 ± 0.02[b] |
| +FeMn | 0.47 ± 0.02[a] | 3.41 ± 1.16[b] | 718 ± 23[b] | 0.37 ± 0.01[a] |

The dark-adapted maximum photosystem II quantum yield ($F_v/F_m$), the functional absorption cross-section of PSII ($\sigma_{PSII}$), the time constant for electron transfer at PSII ($\tau_{QA}$) and the connectivity between adjacent photosystems ($P$) were determined at the start and the end of the experiment from the communities sampled West and East after exposure to different Fe and Mn availabilities. Values represent the mean ± SD ($n > 10$). Different letters indicate significant differences between treatments ($p < 0.05$).

differ, as previously observed for different *Pseudo-nitzschia* species[50]. Unfortunately, almost nothing is known on the inter- and intraspecific TM requirement of different *Fragilariopsis* species.

While the single addition of Fe significantly enhanced growth of the bloom-forming species *P. antarctica* at West, the combined addition of FeMn did surprisingly not trigger its growth (Fig. 3c). While we are unsure how to explain these effects, top-down and bottom-up controls such as growth inhibition and/or competition could be an explanation. The very low dFe concentrations at West (0.05 nM) potentially triggered competition for Fe among the different members of the community[50]. As the bloom-forming diatoms are generally considered as *r* strategists[25], they usually have high resource uptake capabilities, thus providing them with a competitive advantage over other groups. Due to its much higher cell abundance relative to *P. antarctica* at West (Supplementary Table 1), the bloom-forming *Fragilariopsis* sp. may have been more efficient than the prymnesiophyte in taking up the added Fe and Mn, potentially causing the observed diminished growth rates by *P. antarctica* (Fig. 3c). Similar to our findings here, during a diatom-dominated bloom, strong competition for dFe between microbes, small phytoplankton and diatoms was reported[51].

Different to location West, the analysis of the bulk parameter Chl*a* indicated a Fe-Mn co-limitation effect at the community level at East. While the effect of FeMn compared to Fe alone had a limited effect on total Chl*a* (higher by 0.02 d⁻¹), the small cells (< 2 µm) reached highest values only when both Fe and Mn were added (higher by 0.14 d⁻¹, Fig. 4a). A modest, but significant growth stimulation by 0.02 day⁻¹ after addition of Fe and Mn together relative to Fe alone was also observed for the picoeukaryote group P2, which numerically dominated the East community (Fig. 4b) and thus contributed most to the Chl*a* pool (Fig. 4a). In fact, the cell abundance of P2 was enhanced by ~32% in the +FeMn relative to the +Fe treatment (Supplementary Table 2). These results indicate a biochemical dependent co-limitation between Fe and Mn (Type III[27]) of the numerically most abundant phytoplankton group at East. The very low dMn concentration observed at station East (0.08 nM) possibly enhanced the Fe demand for growth of this group, which was only relieved when Fe and Mn were amended together. Thus, due to different cellular Fe and Mn requirements and TMs availability[6], not all members of a community may be affected

equally by co-limitation (also defined as community co-limitation[25]). As a consequence, this can result in regional and interspecific differences in cellular phytoplankton stoichiometries and possibly lead also to a cascade effect of limitation by trace metals other than Fe and Mn[6,24].

Reduced Fe availability usually impacts photosynthesis, as it is essential in many parts of the electron transport chain[16], typically resulting in decreased $F_v/F_m$ and increased $\sigma_{PSII}$ of phytoplankton cells[13,51]. Thus an inverse relationship between $F_v/F_m$ and $\sigma_{PSII}$ has commonly been used as an indicator of Fe limitation in HNLC waters[29,52]. Both initial phytoplankton communities displayed low $F_v/F_m$ values, accompanied by large $\sigma_{PSII}$ (Table 3), indicative of Fe-limited communities compensating for fewer photosynthetic reaction centres[53]. Fe-limited phytoplankton is also known to suffer from higher oxidative stress requiring also more Mn to produce the antioxidant enzyme superoxide dismutase and thereby preventing cell damage[48]. Some studies suggested that low $F_v/F_m$ values may also be associated with low Mn concentrations[21–23,28]. In our study, only the addition of Fe and Mn together led to the highest $F_v/F_m$ values of both final communities (Table 3) indicating that besides Fe, Mn availability also influenced the $F_v/F_m$ signal. Hence, the addition of Mn next to Fe enabled the cells to prevent oxidative stress and to reach maximum photosynthetic efficiency. Based on this and previous studies[21–23,28], one needs to be careful when interpreting low $F_v/F_m$ values detected in the field as they could be the result of multiple TM limitations, in our specific case Fe and Mn.

To assess whether low $F_v/F_m$ values reflect a limitation by multiple TMs, especially Fe and Mn, we sampled 22 contrasting in situ stations across the Drake Passage and along the Western Antarctic Peninsula (Fig. 1; Tables 1, 4). Across all sites, a wide range of dFe (0.05–4.51 nM) and dMn (0.08–6.92 nM) concentrations as well as of $F_v/F_m$ values (0.12–0.56) was found (Tables 3, 4). Considering that $F_v/F_m$ is commonly used as a proxy for Fe limitation in the field[29,52], it can be assumed that low dFe concentrations should be correlated with low $F_v/F_m$ values, something which was, however, not the case across the 22 sampling stations (Fig. 5a). As the sampling occurred in late summer/ early autumn, and coincided with the time in which periodic dust inputs from Patagonia have already been reported[54], this lack of correlation could be potentially explained by these sporadic events. Iron associated to dust can be poorly soluble and not very bioavailable to SO phytoplankton[55], leading thus on the one hand to relatively high dFe concentrations, but on the other hand to low $F_v/F_m$ values. Mn is required in the oxygen-evolving complex[17], and Mn-limited diatoms have previously been reported to have compromised photosynthetic efficiency of PSII, resulting in low $F_v/F_m$ values[48]. In fact, there was a significant positive correlation between $F_v/F_m$ and dMn concentrations across all stations (Fig. 5b). Furthermore, our results indicate that Mn availability also influences the photophysiological signature of Fe-limited phytoplankton in the field. Fe limitation can cause higher oxidative stress and thus increase the Mn-demand of phytoplankton[48]. Therefore, the degree of Fe limitation phytoplankton experienced in the Drake Passage could be enhanced due to the low dMn concentration observed here (0.05–1.21 nM). Together the results from our observational and experimental data, highlight that $F_v/F_m$ may also be a photophysiological signature for a TM co-limitation with Fe, in our case for Fe and Mn. Hence, TM co-limitation by phytoplankton can be difficult to dismantle in the field purely on the basis of low $F_v/F_m$ values alone and can only be confirmed through the performance of in situ TM addition incubation experiments. Clearly, further investigations are needed in order to assess the occurrence of co-limitation by Mn and other trace nutrients together with Fe in the SO.

**Table 4 Photosynthetic efficiency and dissolved trace metals across the Drake Passage and the Western Antarctic Peninsula.**

|  | Latitude | Longitude | dFe | dMn | dFe:dMn | $F_v/F_m$ |
|---|---|---|---|---|---|---|
|  | min$^{-1}$ | min$^{-1}$ | nM | nM | nM:nM | rel.unit |
| *PS97* |  |  |  |  |  |  |
| Station #1 | 60° 01.08′ S | 65° 41.63′ W | 2.88 | 0.36 | 8.0 | 0.17 ± 0.02 |
| Station #2 | 60° 44.73′ S | 65° 2.00′ W | *nd* | 1.52 | *nd* | 0.27 ± 0.01 |
| Station #3 | 62° 29.90′ S | 64° 17.66′ W | 0.50 | 0.39 | 1.28 | 0.21 ± 0.02 |
| Station #4 | 62° 35.03′ S | 59° 41.33′ W | 4.51 | 6.92 | 0.65 | 0.53 ± 0.03 |
| Station #5 | 62° 26.57′ S | 59° 12.24′ W | 2.93 | 5.46 | 0.54 | 0.52 ± 0.02 |
| Station #6 | 62° 25.32′ S | 58° 29.35′ W | 1.53 | 1.53 | 1 | 0.43 ± 0.04 |
| Station #7 | 61° 49.53′ S | 55° 39.02′ W | 0.89 | 1.27 | 0.7 | 0.45 ± 0.01 |
| Station #8 | 60° 29.94′ S | 55° 29.70′ W | 2.99 | 1.21 | 2.4 | 0.36 ± 0.02 |
| Station #9 | 58° 5.01′ S | 66° 3.00′ W | 3.21 | 0.31 | 10.3 | 0.12 ± 0.03 |
| *PS112* |  |  |  |  |  |  |
| Station #10 | 62° 45.60′ S | 59° 00.42′ W | 1.85 | 2.51 | 0.74 | 0.56 ± 0.01 |
| Station #11 | 63° 00.00′ S | 60° 30.00′ W | 3.76 | 2.62 | 1.43 | 0.36 ± 0.01 |
| Station #12 | 62° 53.16′ S | 60° 14.32′ W | 2.42 | 3.05 | 0.79 | 0.52 ± 0.03 |
| Station #13 | 62° 14.23′ S | 64° 33.41′ W | 4.44 | 0.08 | 55.5 | 0.19 ± 0.02 |
| Station #14 | 61° 45.08′ S | 62° 00.49′ W | 1.31 | 0.42 | 3.12 | 0.31 ± 0.01 |
| Station #15 | 61° 01.33′ S | 54° 47.30′ W | 3.09 | 3.21 | 0.96 | 0.32 ± 0.01 |
| Station #16 | 60° 41.80′ S | 54° 34.24′ W | 0.29 | 0.98 | 0.29 | 0.35 ± 0.01 |
| Station #17 | 63° 38.78′ S | 56° 29.57′ W | 3.13 | 3.60 | 0.87 | 0.21 ± 0.01 |
| Station #18 | 61° 59.25′ S | 53° 59.37′ W | 1.92 | 1.85 | 1.04 | 0.39 ± 0.01 |
| Station #19 | 60° 57.51′ S | 54° 44.69′ W | 1.31 | 2.18 | 0.61 | 0.41 ± *nd* |
| Station #20 | 58° 35.64′ S | 58° 61.37′ W | 0.06 | 0.11 | 0.54 | 0.12 ± 0.01 |

Concentrations of total dissolved iron (dFe), dissolved manganese (dMn) and the 1h-dark-adapted maximum photosystem II quantum yield ($F_v/F_m$) were determined for 20 in situ stations sampled during the Polarstern expeditions PS97 and PS112 using GoFlo bottles. Values represent the mean ± SD ($n = 3$). *nd* denotes not determined.

Although light and grazing by microzooplankton are important factors[56], since 1990[7] it has been demonstrated that Fe is the predominant driver controlling phytoplankton productivity in HNLC regions. Recent studies highlighted that in addition to Fe other micro-nutrients such as Mn, cobalt, zinc and vitamin $B_{12}$ can also influence phytoplankton biomass. This makes the investigations, to which degree phytoplankton cells in HNLC waters experience co-limitation of trace nutrients other than Fe, a growing field of study[12,21,22,28,47]. Until now, most experiments focused on the single addition of Fe and only three studies tested the effect of Mn alone[19] and/or in combination with Fe[22,23]. However, the latter two studies looked either at proteomic responses or at the total biomass response, but they could not resolve any Fe-Mn induced changes in species composition. Our study is the first to show that even though most members were primarily Fe-limited within two SO phytoplankton communities a dominant phytoplankton group, however, was identified to be limited by both Fe and Mn. The reason for this could be divergent trace metal requirements among phytoplankton species and groups[12,27,57] and Fe and Mn availabilities. As only few FeMn-enrichment experiments were performed in SO waters so far[22,23], the occurrence of FeMn co-limitation of phytoplankton communities across the SO has not yet thoroughly been assessed. Therefore, one could speculate that within an Fe-limited phytoplankton community some phytoplankton species or groups could be still Fe-Mn co-limited, remaining, however, yet undetected due to a lack of experiments and thus little data availability. In this study, we also observed a significant positive correlation between dMn concentrations and $F_v/F_m$ values at 22 in situ sampling stations across the Drake Passage and the Western Antarctic Peninsula region, something which is usually associated with dFe concentrations[7–9]. In the future, the SO will experience drastic climatic changes, with stronger westerly winds[58] causing a deepening of the upper mixed layer, thereby increasing Fe input from deeper waters and decreasing light availability. The consequences of these two opposite drivers (more Fe, less light) on phytoplankton productivity are not yet clear. The comparison of various climate models indeed suggests that deeper mixing could bring up additional Fe and stimulate biological production[59]. As a consequence, the Fe stress of HNLC regions such as the Drake Passage could be relieved, thereby uncovering potential Mn co-limitation effects. Therefore, dFe must not be the only TM considered when assessing future biogeochemical changes in this climate-relevant region of the world ocean.

## Methods

**Water collection and experimental set-up.** Two shipboard bottle incubation experiments were conducted to study the potential concomitant effects of Fe and Mn addition on SO phytoplankton community structure at two locations of the Drake Passage during the RV Polarstern expedition PS97 in 2016. At the first location, HNLC seawater was collected on 1 March 2016 at 60° 24.78′ S/66° 21.85′ W at 25 m depth (BIO 1, referred to as Station West hereafter, Fig. 1). In the eastern Drake Passage at 58° 52.17′ S/60° 51.92′ W, HNLC seawater was sampled on 17 March 2016 at 25 m depth (BIO 3, referred to as Station East hereafter, Fig. 1). Two companion studies have been performed at the same locations with a more detailed focus on how the presence of different organic ligands influenced Fe uptake rates and trace metal chemistry[60] as well as phytoplankton, microbial and viral abundances[35]. To minimize contamination, trace metal clean (TMC) techniques (sampling and handling) based on GEOTRACES guidelines[61] were used. Hence, all tubing, reservoir carboys, incubation bottles, and other equipment were acid-cleaned prior to the cruise using TMC techniques. Briefly, they had been sequentially soaked for 1 week in 1% Citranox and for 1 week in 1.2 mol L$^{-1}$ hydrochloric acid (PA grade, Merck Millipore Corporation, Darmstadt, Germany). Between each soaking step, the bottles were rinsed seven times with ultrapure water (Merck Millipore Corporation, Darmstadt, Germany). Finally, the TMC equipment/bottles were air-dried under a clean bench (US class 100, Opta, Bensheim, Germany) and packed in three polyethylene (PE) bags for storage. At the two stations, to avoid possible contamination from the ship, HNLC seawater was pumped from 25 m directly into the clean laboratory container (US class 100, Opta, Bensheim, Germany) using a Teflon membrane pump (Almatec, Futur 50) (Table 1). Before each sampling, the pump and hosing were flushed for 1 h with seawater, ensuring that the system was well rinsed. In the clean container, acid-cleaned 2.5 L polycarbonate (PC) bottles were filled with HNLC water inside an extra laminar flow hood after having passed through a cleaned 200 μm mesh, removing large grazers. The mesh was visually inspected, rinsed and stored in 1 M HCl between stations.

The Control treatment was the sampled HNLC seawater without any TM addition while the other three treatments were enriched with either FeCl$_3$ alone (0.5 nmol L$^{-1}$, AAS standard, TraceCERT, Fluka; +Fe treatment) or MnCl$_2$ alone (1 nmol L$^{-1}$, AAS standard, Trace-CERT, Fluka; +Mn treatment) or both trace

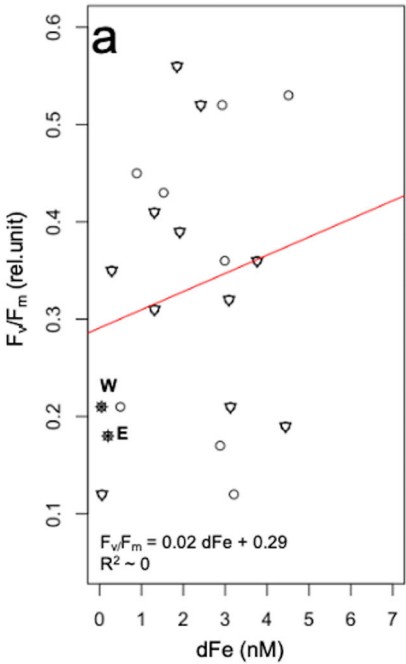
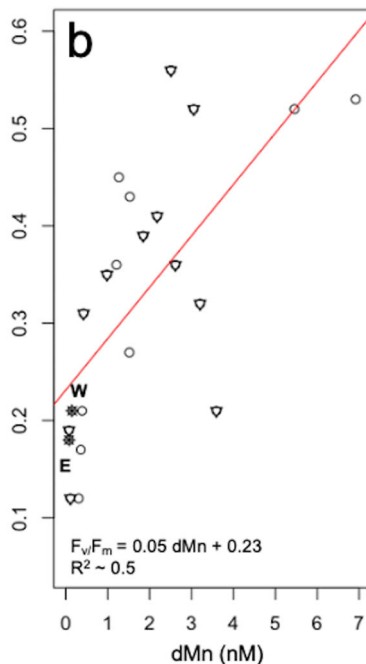

**Fig. 5 Relation between the photosynthetic efficiency ($F_v/F_m$) and dissolved trace metals (dFe and dMn).** Linear regression (red line) between $F_v/F_m$ and **a** the concentrations of total dissolved iron (dFe) or **b** the concentrations of total dissolved manganese (dMn). The data were determined at 22 in situ stations sampled across the Drake Passage and around the Western Antarctic Peninsula, including BIO1 as station West and BIO3 as station East (based on data of Tables 1, 3, 4). The two experimental stations West and East are indicated by dark circles while the PS97 stations are denoted by white circles and the PS112 stations by white triangles.

metals together (+FeMn treatment). All treatments were done in triplicate 2.5 L PC bottles. No ethylenediaminetetraacetic acid (EDTA) was added in order to avoid alteration of the natural seawater trace metal chemistry[62]. Low dFe and dMn additions were made in order to avoid formation of inorganic colloids in our experiments. Fe chemical speciation data from a joint study[60] during PS97 demonstrated that in situ ligands were present in sufficient quantity to react with the $FeCl_3$ added. No additional macronutrients were added to the incubation bottles (Table 1). As in previous Fe addition experiments, final sampling took place between 7 and 15 days depending on the treatment, with Fe enriched treatment harvested first[55,63]. In our study, the duration of both experiments ranged between 14 and 17 days according to the respective treatment. The photosynthetic efficiency of the community was assessed every 2–4 days after 1 h dark-acclimation via a Fast Repetition Rate Fluorometer (FRRf, FastOcean PTX sensor, Chelsea Technologies Group (CTG) Ltd, West Molesey, UK) at 1 °C (Supplementary Fig. 3). All incubation bottles were maintained at 30 µmol photons $m^{-2} s^{-1}$ white light (providing the full light spectrum) under a 16:8 (light:dark) hour cycle at 1 °C, mimicking natural conditions typical for March of this region. To avoid sedimentation of the cells, all incubation bottles were gently rotated every day. At each sampling location, at the start and the end of each experiment, samples were taken for determination of seawater chemistry, phytoplankton community composition, elemental composition and photophysiology (see below). In parallel of the two Fe-Mn enrichment experiments, 9 in situ stations were sampled trace metal clean (TMC) (Fig. 1) using GoFlo bottles to determine concentrations of total dissolved Fe (dFe) and Mn (dMn) of the collected seawater. In addition, the maximum photosystem II quantum yield ($F_v/F_m$) was determined after 1 h of dark-acclimation using the FRRF.

**Seawater chemistry**. dFe and dMn concentrations were estimated from the initially sampled seawater. To this end, 100 mL of seawater were filtered through HCl-cleaned polycarbonate filters (0.2 µm pore size) using a TMC Nalgene filtration system and the filtrate was collected into PE bottle and stored triple bagged at 2 °C until analysis. Concentrations of the dFe and dMn were determined on a SeaFast system (Elemental Scientific, Omaha, NE, USA)[64,65] coupled to an inductively coupled plasma mass spectrometer (ICP-MS, Element2, Thermo Fisher Scientific, resolution of $R = 2000$). During the pre-concentration step, an imino-diacetate (IDA) chelation column (part number CF-N-0200, Elemental Scientific) was used. The pre-filtered seawater samples were acidified to pH = 1.7 with a double-distilled nitric acid ($HNO_3$) and were UV-treated using a 450 W photo-chemical UV power supply (ACE GLASS Inc., Vineland, NJ, USA) to minimize adsorption of TMs onto the bottle walls and to reduce the formation of Mn and Fe hydroxides during storage. During each UV digestion step, two blanks were taken. The ICP-MS was optimized daily to achieve oxide-forming rates below 0.3%. Each

seawater sample was analysed via standard addition to minimize any matrix effects, which might influence the quality of the analysis. To assess the accuracy and precision of the method, a NASS-7 (National Research Council of Canada) reference standard was analysed in a 1:10 dilution (corresponding to environmentally representative concentrations) at the beginning, in the middle and at the end of each run (two batch runs; $n = 18$). The measured values were in the limits of the certified NASS-7 reference material, with a concentration of $351 \pm 26$ ng $L^{-1}$ for dFe and $750 \pm 60$ ng $L^{-1}$ for dMn (mean ± standard deviation). The detection limits for Mn and Fe were 8.1 pM and 81.8 pM, respectively. The dissolved macronutrient concentrations (total nitrate (nitrite + nitrate), phosphate and silicate) were determined colorimetrically in the laboratory on a QuAAtro autoanalyzer (Seal Analyticals).

**Phytoplankton community characterization by light microscopy**. To determine the effects of the different treatments on the microplankton composition for the two Fe-Mn enrichment experiments, unfiltered seawater was collected at the start and the end of both experiments for later analysis via light microscopy in the home laboratory. Briefly, samples were fixed with hexamine-buffered formalin solution (2% final concentration) and Lugol's solution (1% final concentration) and stored at 2 °C in the dark until taxonomic analysis. All samples were allowed to settle in Utermöhl sedimentation chambers (Hydrobios, Altenholz, Germany) for at least 24 h and were analysed on an inverted light microscope (Axiovert 200; Zeiss), according to the method of Utermöhl[66], following the recommendations of Edler[67]. Species were counted and identified according to taxonomic literature[68]. Each aliquot was examined until at least 400 cells had been counted.

**Phytoplankton community characterization by flow cytometry**. Autotrophic pico- and nanoeukaryotes as well as heterotrophic bacteria were analysed via flow cytometry. At the start and the end of the experiment, samples were preserved with 10% buffered formalin, flash-frozen in liquid nitrogen, and analysed flow cyto-metrically to assess picoplankton densities[69]. Abundances of heterotrophic bacteria (stained with Synergy Brands [SYBRTM] Green I), phycoerythrin-containing picocyanobacteria, and photosynthetic picoeukaryotes were determined by means of a BD Accuri™ C6 Plus flow cytometer (Becton, Dickinson and Company) using fluorescence patterns and particle size from side angle light scatter[69,70]. Before running the samples, 2 µL beads (Sperotech - Rainbow Fluorescent Particles (RFPs) - 2.11 µm) were added to each treatment as a size and fluorescence reference. Then pico- (P), nanoeukaryotes and (N) were identified based on side scatter versus FL-3 and heterotrophic bacteria (B) on side scatter versus FL-1. Three P subgroups (0.2–2 µm) were differentiated according to their size: small (P1), medium (P2) and large (P3), according to sub-cluster of events, as shown in the Supplementary Fig. 1.

Based on cell abundances from microscopy and flow cytometry, specific growth rate per day (μ, d$^{-1}$), was calculated for all cells using:

$$\mu = \frac{ln(Nt_2 - Nt_1)}{dt}, \quad (1)$$

where $N_{t1}$ and $N_{t2}$ are the cell abundances at the start and at the end, respectively, of each incubation experiment, while $dt$ denotes the incubation time in days.

**Chlorophyll a content**. Chlorophyll a (Chla) samples were taken at the beginning and the end of both experiments. In order to compare the contribution of large (> 2 μm) relative to small cells (0.2–2 μm), on average 1000 ml at the beginning and 250 mL at the end of the experiment seawater were filtered onto 0.2 μm and 2 μm polycarbonate filters (Whatman, Wisconsin, USA). All samples were directly flash-frozen into liquid nitrogen (N$_2$) and then stored at −80 °C in the dark until further analysis. After being homogenized, samples were extracted in 90% acetone for 24 h at 4 °C in the dark and analysed fluorometrically[71] on a Trilogy Fluorometer (Turner Design, San Jose, CA, USA) using the non-acidification module. Based on Chla content of the cells, the specific growth rate per day (μ, d$^{-1}$), was calculated from Eq. (1) for the two different size fractions (small and large).

**Chlorophyll a fluorescence**. At the start, during and at the end of both experiments as well as from the 9 stations, chlorophyll a fluorescence measurements were collected using a Fast Repetition Rate Fluorometer (FRRf) coupled to a FastAct Laboratory system (FastOcean PTX), both from Chelsea Technologies Group. The excitation wavelengths of the fluorometer's LEDs were 450 nm, 530 nm and 624 nm and the light intensity was automatically adjusted between 0.66 and $1.2 \times 10^{22}$ photons m$^{-2}$ s$^{-1}$. The single turnover mode was set with a saturation phase of 100 flashlets on a 2 μs pitch followed by a relaxing phase of 40 flashlets on a 50 μs pitch. In order to calculate the maximum quantum yield of photosystem II (PSII) (F$_v$/F$_m$ [rel. unit]), the minimum (F$_0$) and maximum (F$_m$), chlorophyll a fluorescence of PSII were determined after 1 h of dark acclimation.

$$\frac{F_v}{F_m} = (F_m - F_0)/F_m, \quad (2)$$

From Oxborough et al.[72], the functional absorption cross-section of PSII (σ$_{PSII}$, nm$^2$ PSII$^{-1}$), the time constant for electron transport at the acceptor side of PSII (τ$_{Qa}$, μs) and the connectivity factor (P, dimensionless) were derived using the FastPro8 Software (Version 1.0.55, Kevin Oxborough, CTG Ltd).

**Statistics and reproducibility**. To test the normal distribution and equal variances of the datasets, Shapiro–Wilk tests were performed. A one-way analysis of variance (ANOVA) was conducted in order to assess the impact of Fe and Mn availability compared to the Control treatment on cell abundance and Chla accumulation rate. As post hoc tests, the Tukey honest significant difference (HSD) test and additional pairwise t-tests such as the Benjamini–Hochberg method were used between the mean group for a pairwise comparison of the effect of the two factors. A $p < 0.05$ was used to establish significant differences among treatments compared to the control. Linear regression and Pearson correlation test were used to test the dependence and correlation between two continuous variables, with $R^2$, being the coefficient of determination and $r$, being the correlation coefficient. All statistical analyses were performed with R Studio (version 1.1.463, © 2009–2016) and all maps with Ocean Data View[73]. R packages to reproduce statistical analysis are stats (aov, lm and cor.test functions), ggplot2 (ggplot function) and agricolae (HSD.test function). The Shannon diversity index ($H$) was calculated from:

$$H = \sum_{i=1}^{R} pi \ln pi, \quad (3)$$

where $R$ is the total number of species and $pi$ the relative abundance of species.

**Reporting summary**. Further information on research design is available in the Nature Research Reporting Summary linked to this article.

## Data availability
All data needed to evaluate the conclusions in the paper are present in the paper and/or Supplementary Materials and are freely available from the PANGAEA data repository: https://doi.org/10.1594/PANGAEA.935868.

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

## Acknowledgements

J.B. was funded by the Deutsche Forschungsgemeinschaft (DFG) in the framework of the priority programme'Antarctic Research with comparative investigations in Arctic ice areas', project TR 899/4-1. F.K. was also supported by the Deutsche For-schungsgemeinschaft (SPP1158, KO5563/1-1'ViTMeD'). Finally, we would like to thank the captain and crew of RV'Polarstern' during PS97.

## Author contributions

S.T. and F.K. designed and conducted the experiments and measured the samples of the experiments. J.B. and F.K. analysed the data; J.B., S.T., F.K. and C.H. wrote the manuscript.

## Funding

## Competing interests

The authors declare no competing interests.
