## [Transparent Peer Review File · Communications Biology]

Reviewers' comments:

Reviewer #1 (Remarks to the Author):

Review of Balaguer et al.

This paper tackles the question of the role of iron and manganese in shaping phytoplankton activity and community composition from a set of experiments conducted in Drake Passage.

The results suggest evidence of Fe-Mn colimitation and an inhibitory effect of Mn addition on *Phaeocystis*. These results emerge from a set of standard bioassay experiments at two locations, for which phytoplankton community composition and photophysiology (Fv/Fm, sigmaPSII) were measured.

Sampling at a set of additional stations revealed no link between dFe and Fv/Fm, while there was a positive correlation between Fv/Fm and dMn, and a negative correlation with dFe:dMn.

Right now, I do not think the paper is ready for publication in *Communications Biology*. It reads too much like a report, without sufficient mechanistic insight for a broad audience. There is a lot of description of the result associated with different phytoplankton groups and a set of photophysiological measurements, but we are never provided with much detail on mechanisms and processes. The new understanding of how the Southern Ocean system operates is, right now, not there. I was hoping for this to emerge in the discussion section, but this ended up largely being a restating of the results.

For instance, I do not think it is new knowledge to state that the Southern Ocean is a region with high potential for Fe-Mn colimitation. As the authors know, this was first proposed by Martin decades ago.

The link between Fe/Mn additions and the response of different phytoplankton groups and bacteria is interesting, but it is again a descriptive part of the paper with limited mechanistic insight. For example, the authors state that the low response from bacteria is because organic carbon is limiting, but was that measured during the experiments? Equally, why do the authors think different groups are responding differently to additions of Fe, Mn and both Fe and Mn? There is some (very brief) discussion of surface area to volume, but otherwise I was left unclear as to what was driving the results? This part of the paper should be strengthened significantly.

Lastly, the links between photophysiological measurements like Fv/Fm and some bulk micronutrient measurements are also not elaborated. This is an intriguing results, but the discussion is also underwhelming as it spends too much time restating results. I was left confused as to why dMn should correlate with Fv/Fm and I question whether this is at all causal. I was hoping / expecting that this part could be expanded with a wider Southern Ocean dataset.

Reviewer #2 (Remarks to the Author):

Brief Summary of the manuscript:

Balaguer and co-authors present evidence for phytoplankton species composition control by both iron (Fe) and manganese (Mn) in the Southern Ocean. Results come from shipboard bottle incubation experiments conducted in the Drake Passage where Fe and Mn were added alone and in combination. Authors further show that specific phytoplankton groups found in the Drake Passage such as the diatom *Fragilariopsis* were Fe-Mn co-limited, while other groups were only Fe-limited. Ultimately, these results support the availability of multiple trace metals being a major control for phytoplankton community growth and structure and not only Fe. These are some of the first natural (whole) phytoplankton community experimental results that show that the relief of Fe-Mn co-limitation in some parts of the Southern Ocean, recently shown by Browning et al. (2021), not only promotes total

community growth and biomass increase but also responses from specific phytoplankton groups/species.

Overall impression of the work:

I think the results presented in this manuscript are very interesting and an important contribution to the current state of knowledge regarding the role of low trace metal concentrations in the Southern Ocean. This in combination with the evidence from Browning et al., 2021 will likely bring about many new studies and objectives for research cruises and subsequent evaluation of trace metal parameters in current ocean biogeochemical models. I applaud the combined use of observational results from other in situ stations with that from the two experiments to provide better context.

However, some parts of the manuscript read difficult and do not flow smoothly. For example, a few abbreviations are not defined when first used. Although not essential, authors should consider reducing the use of abbreviations that are not used often. The visual presentation of results could be improved with both edits to existing figures (e.g. adding colour to bar plots). Additions to the method section is also suggested to both improve readability and ensure common queries/concerns of bottle incubation experiments are addressed. Also, due to the structure/format of this journal (methods at the end), a short description of each variable could be useful when first discussed in results to improve readability/context and better inform the reader. There are a few statements in the discussion that are not properly explained and/or unclear. Specifically, I suggest to further address in the discussion the likely reasons that could have led to experiment results indicating that Phaeocystis was inhibited by Mn. Brief mentions were already made to the uncertainty for why Phaeocystis appears to be inhibited. If this is the case, I would suggest rephrasing Line 20 in the abstract to be less direct. Below are specific comments and suggestions.

Introduction:

Line 30: Rephrase to "The Southern Ocean is unique compared to much of the global ocean as it accounts for over 40% uptake of anthropogenic carbon dioxide (CO₂), making it the largest oceanic CO₂ sink (1,2)" If space allows, add one sentence on what role phytoplankton plays in this uptake and add Buesseler et al. (2020) as a reference who recently reported that the role of phytoplankton and quantity of carbon sequestration occurring in the Southern Ocean is underestimated.

Line 31: Note first use of "CO₂". Please define in text here.

Line 32: Define "SO" abbreviation earlier in Line 30. Also rephrase to "...SO (3), which results in the SO being characterized as the world's largest..."

Line 33: Suggest adding short sentence before "Low essential trace metals..." about trace metals as important micronutrients to phytoplankton with a reference such as Twining & Baines (2013). Currently the use of "essential" is unclear. Also suggest to rephrase to: "Low concentrations of essential trace metals (such as iron,...".

Line 36: Change "biomass build up" to "growth".

Line 37: Change "different" to "various"

Line 41: Change "trace metal needed in the..." to "trace metal needed during photosynthesis in the..."

Line 43: Expand on why the scavenging of ROS is important.

Line 45: Change "Just like Fe," to "Similar to Fe,"

Line 47-48: Be specific on what type of studies is referred to here: observational or experimental. Maybe rephrase to "Several studies have subsequently suggested the occurrence of Mn co-limitation with Fe of SO phytoplankton".

Line 55: Not sure if it is necessary to point out here in the introduction what form of Fe "FeCl₃" was used in additions.

Line 58: Add "the" before "Drake Passage"

Line 58: Is there a reference on the time it takes for a community change after a change in their environment?

Line 67: Change "shapes" to singular "shape"

Line 73: Change to "... and picoeukaryotes in natural communities sampled at two different locations..."

Since the results from this study might also lead to discussions about the single and/or co-limitation of other metals as well, it is worth briefly mentioning previous work that discussed/suggested co-limitation of multiple metals other than Fe if length restrictions allow.

Results:

Line 80: Make sure the abbreviations "Chla, dFe and dMn" are defined at first use. For example: "dissolved iron (dFe)". Also review units of Chla in Table 1. Chla units in mg/L. 0.09mg/L = 90ug/L. This seems unlikely. Please explain or revise.

Line 93: be more specific here and state that picoeukaryotes were determined via flow cytometry. At this stage of the manuscript, it is unclear why picoeukaryotes are defined and grouped separately from the microphytoplankton. Perhaps state briefly when first discussed how each was determined. Also, define here the differences between the various P groups.

Line 95 to 96: Again, briefly mention how N and B were determined. Perhaps revise the first sentence of the paragraph at Line 93 to state that Flow cytometry analysis revealed that P, N and B were present at both locations.

Line 101: "nitrite/nitrate" Is this the sum of nitrate and nitrite or just nitrate. See the method section for further comments. In tables, NO_x is defined but not in text as far as I can see. At first use of Nitrate in results define that it refers to the sum of nitrite and nitrate.

Line 104: Define here how chla accumulation rates are calculated. Especially since detailed methods are at end for this journal structure. It might provide better context to the non-expert reader if similar figures to Fig2 and Fig3 was added to the supplemental showing the absolute chla concentrations and cell abundances at the beginning and end of each treatment and control for both experiments.

Line 112: Refer to earlier comment relating to defining the P groups. The previous suggestion to add figures showing the absolute abundances of each method would especially be useful to gain a better perspective on the negative growth rates shown. Again, briefly define/explain here how growth rate is determined with flow cytometry results or provide a detailed explanation of this in the methods section and refer to the specific method section here. This applies to all growth/accumulation rates used.

Line 145: Visually it seems chaetoceros growth decreased at +Mn treatment compared to control? Please confirm.

Line 155: Depending on the length, I would suggest keeping the same structure as previous sections: "Experiment West", "Experiment East" and then the additional section here "in situ stations".

Line 158 and photophysiological results in general: Reporting summary states "The physiological health of the community was assessed every 2-4 days and enable to determine the day of the harvesting (=sampling)." However, this is not clearly stated in the methods of Fv/Fm (Chlorophyll a fluorescence). If this is the case, an additional figure of Fv/Fm and SigmaPSII over time in the supplemental would be beneficial. This could be used to briefly discuss photophysiological status overtime after additions that due to the length of the experiment the Fv/Fm at the end of the treatment might have changed during the experiment. Viljoen et al. (2018) showed a rapid increase in Fv/Fm within one day after Fe addition treatment of Antarctic waters whereafter Fv/Fm reached a plateau after 2-3days and then began to decrease.

Discussion:

Line 192: Change "macronutrients" to singular "macronutrient". I like Table 1 as it provides a good overview of initial conditions. However, authors should consider adding the initial Fv/Fm values for each experiment from Table 4 to Table 1 as well.

Line 193-195: Sentence is a bit unclear. Were dMn concentrations of this study lower than what was measured by others for the Drake Passage? Or was dMn below 0.2nM, similar to previous studies.

Line 196: Change "indicative for" to "indicative of". Change "In line with this," to "Accordingly,"

Line 200: Change "predominant" to "dominant"

Line 203: Insert "low" before "biomass, plankton abundances and ..."

Line 214-215: A reference to the earlier suggested supplemental absolute abundance figures (Fluorometer Chla, Flow cytometry and microscope determined microphytoplankton) would be useful here to provide better context for the total abundance reached for the Fe only addition treatment and others.

Line 222-223: Rephrase sentence starting "Since the addition..." to "The addition of Fe led to similar responses as the addition of both trace metals together, which indicates that Fe..."

Line 224-226: Statement a bit unclear. It is unclear to me how the stimulation in growth of the two P groups due to Fe addition show how small cells better cope with low Fe. I suggest rephrasing in some

way to improve clarity.

Line 230-231: Rephrase sentence starting "This result clearly shows..." to "Therefore, the diatom *Fragilariopsis* sp. was clearly co-limited by both Fe and Mn."

Line 237: Add "an increase in" after "stimulated"

Line 239: Change "...and was attributed to..." to "was mostly attributed to"

Line 241: This is an interesting point. I might have missed it in a later section, but I suggest expanding here or in the dFe:dMn ratio section later on the possible reasons for why P2 is co-limited at East where dMn is much lower than West where *Fragilariopsis* was only species/group co-limited.

Line 252: Explain what is meant by "competition". Are the authors referring to the competition between species/groups or metal competition? Expand on why the decrease in *Phaeocystis* growth is more pronounced in East experiment than West. Include possibilities of pre-acclimation to certain initial conditions that might have led to this result. If related to the dFe:dMn ratios discussed later, refer the reader to that section. Refer to a combination of metal availabilities at these stations, i.e. what species have absolute requirements vs can substitute Mn for another metal. This can be introduced in introduction by referring to Twining & Baines (2013), observational study Viljoen et al. (2019) and experimental study of Browning et al (2014) for the possibility of multiple nutrient limitations. Also, some studies already referenced here show the importance of Mn as an absolute requirement for most phytoplankton. This could possibly be used here as support for Mn not inhibiting *Phaeocystis* growth, but rather the competition by dominant diatoms for finite nutrient resources in bottle incubation experiments.

Are there initial or in situ concentrations for other metals available? Or nutrient concentrations at the end of experiments to see possibility of other limiting macro- and micronutrients?

Line 253: Reference Twining & Baines 2013 at end of sentence.

Line 255: Fv/Fm discussion section: Currently the connection between Fe availability and Fv/Fm needs to be expanded. First sentence is a nice overall intro to this section. However, after this a short example of how Fe impacts Fv/Fm and why would link the first sentence to the rest of the paragraph better. Or expand first sentence to include (remind reader) how Fv/Fm fits into this statement. One suggestion: change Line 257 to "... transport chain (14), and typically results in decreased Fv/Fm and increased PSII (strzepek 2012; Behrenfeld and miligan, 2013)"

Line 258-259: Comment on why West shows a significant difference in both Fv/Fm and SigmaPSII and East only Fv/Fm in Table 4? Author's state East likely more Mn co limited than West. Could it be due to higher percentage *Fragilariopsis* diatoms in West than in East?

Line 277: Sentence started with symbol/abbreviation "Fe". Rather start sentence with "Iron".

Line 289: Add "to be" after "expected"

Line 292: Rephrase to For example: "However, due to the wide range of dFe:dMn ratios observed for both coastal and open stations (Fig. 4c), one cannot attribute Mn co-limitation to specific marine areas for this study." Add mention of how this observation/statement compares to what Browning observed for coastal vs open. Is this in contradiction or support their observations? Perhaps use this to further support the next sentence.

Line 297: Mention here or in introduction what the other less major drivers could be for the occurrence of HNLC zones.

Line 309: Change "metals" to singular "metal".

Line 315: It was suggested in text earlier that there are other reasons why *P. antarctica* could have appeared to be inhibited in the Mn only experiment. Refer to previous comments. Studies have shown that *phaeocystis* do have a requirement for Mn next to Fe such as for ROS enzyme (Twining & Baines 2013). Maybe expand on possibility for other metal/resource competition, towards the end of such a long experiment.

Line 332: End current sentence after "structure" here and start another sentence "Therefore, dFe must not be the only TM considered..."

Methods:

Line 354: Mention why sampling depth was set to be 25m. Reduce contamination possibility from ship in shallow waters? Logistical reasons?

Line 377: Many other studies did dFe additions of 1nM or more (e.g. Viljoen et al. 2018). Browning et al. 2021 even had 2nM for both dFe and dMn. Please explain and justify why these concentrations

were chosen with reference to other studies.

Line 383: Add here whether bottles were mixed every few days or not and why. Was this maybe done at each FRRf measurement?

Line 384: Add details here on the light controlling equipment used and if known, the applicability of the emitted light spectrum for phytoplankton growth.

Line 385: Add further detail on the variation in the length of each experiment/treatment and what the reasons were.

Line 405: Define "HNO₃"

Line 416-417: Define here that total nitrate (nitrite+nitrate) is used in results. Why is only silicate referred to as "dissolved silicate"? If macronutrient concentrations at the end of experiments are available, it could be useful to calculate nutrient drawdown rates during the experiment and compare to the various growth rates.

Line 420: Add to Light microscopy and Flow cytometry method sections how growth rates were calculated for these methods/results.

Line 443: If possible, define the size range for each P subgroup.

Line 446: Add to the Chlorophyll-a method section the volume filtered for the fluorometer analysis.

Line 487: Mention specific R packages used and their reference (main contributors) or where specific code used can be found if uploaded to a repository.

Line 497: Give references to specific R packages used during statistical analysis to give recognition to its contributors and for readers to know what packages they can use.

Figures & tables:

Figure 1: A more informative and easier to read figure of the study area would be beneficial. For example, increase size or contrast of sample/experiment locations. Add data such as initial community Chla concentration for each location. Perhaps also add similar map figures as supplemental to visually show initial metal concentrations. Increase domain of map north and east. Include "West" and "East" notations on the map to visually remind the reader of how experiments are referred to in the text.

Figures 2 & 3: Suggest to add colour to improve the contrast of error bars for readers to visually see the comparison between treatments. Error bars are not visible for the black bars (e.g. Total, P3 & Phaeocystis).

Table 2 could be turned into figure format to visually represent the contribution of each species/group for each location's initial community. For example, 4x pie charts (2 for microphytoplankton and 2 for picoeukaryotes) or in one barplot format. Essential to ensure that readers are properly aware of the initial community and/or can easily find it when just scanning through. The original table 2 could move to supplemental.

As said in the main text comments, I suggest adding Fv/Fm values to Table 1.

New References not in current manuscript:

Buesseler, K.O., Boyd, P.W., Black, E.E., Siegel, D.A., 2020. Metrics that matter for assessing the ocean biological carbon pump. *Proc. Natl. Acad. Sci. U. S. A.*

<https://doi.org/10.1073/pnas.1918114117>

Viljoen, J.J., Philibert, R., Van Horsten, N., Mtshali, T., Roychoudhury, A.N., Thomalla, S., Fietz, S., 2018. Phytoplankton response in growth, photophysiology and community structure to iron and light in the Polar Frontal Zone and Antarctic waters. *Deep. Res. Part I Oceanogr. Res. Pap.* 141, 118–129.

<https://doi.org/10.1016/j.dsr.2018.09.006>

Viljoen, J.J., Weir, I., Fietz, S., Cloete, R., Looek, J., Philibert, R., Roychoudhury, A.N., 2019. Links between the phytoplankton community composition and trace metal distribution in summer surface waters of the Atlantic Southern Ocean. *Front. Mar. Sci.* 6, 295.

<https://doi.org/10.3389/fmars.2019.00295>

Kind regards,
Johannes J. Viljoen

Reviewer #3 (Remarks to the Author):

General comments:

Balaguer and coauthors present results from two bottle incubation studies looking at the effects of manganese and iron additions in the Southern Ocean. To date, most bottle incubation studies conducted with iron and manganese only examine bulk community-level properties (e.g. total particulate carbon, total chl-a), while these authors present a more fine-grained analysis of taxonomic composition. While I think these results are interesting and the data are valuable, I have major reservations about the analyses conducted and the strength of the conclusions, and even the principal statement that there is a strong effect of Fe-Mn colimitation. There are also some issues with the statistical analysis and methods (see below) that should be addressed. However, I do think with major changes to the manuscript this will be a valuable contribution to the literature.

My first reservation is conceptual. The authors state that Mn-Fe (together) is a “major control” on growth of SO phytoplankton. But they are only highlighting when effects of one experiment is significant, regardless if that outcome is mirrored in the second experiment. Presumably if something is a major controller of SO phytoplankton, there should be some replicability when two experiments are conducted in the same region of the ocean? For example, to me the result that *Fragilariopsis* was Mn-Fe colimited in one experiment but not the other should indicate that Mn-Fe colimitation is actually not very common. I suggest limiting strong statements about conclusions only to those effects that were mirrored between the two experiments.

Another major reservation I have is with respect to the statistical analysis and data presentation (additional details below). There is a lot of emphasis on statistical significance, but the effect size is not emphasized. I would encourage the authors to state the effect size not in percent change, but rather the actual magnitudes of the effect in their appropriate units. I think this might significantly shift the main message of the paper, because the effect sizes comparing +Fe and +FeMn look quite small. Actually, when I first looked at the figures without reading the text I concluded the opposite that +Fe alone is the dominant effect, except in a small number of instances!

Lastly, I did not see a data availability statement. I think depositing the raw data in a public repository like Dryad should be done to maximize community benefit from these difficult-to-collect data!

This is not a reservation, but a thought. It’s interesting that in some cases the experiments had shifted community structure, but the bulk metrics look similar. I think this is an important contribution, since most bottle incubations looking at bulk properties would completely miss that! It might be interesting for the authors to expand on this point in their discussion.

Overarching comments

Figures should show raw data points, not bars +/- the mean. There is a lot of literature on the benefits of presenting data this way, and bar plots are not as transparent as showing the raw data.

Figures showing statistical output of a Tukey’s test should use the convention of a, b, etc. The many symbols are relatively confusing, and it’s not clear with the *, # symbols.

It’s confusing to me why some taxa decreased in abundance from the initial time point. There is a brief sentence on this in the discussion, but I think that should be expanded a bit more. Is there some role for physiological acclimation, i.e. a change of chl-a per cell? How might that influence your results?

Line by line comments

Line 102: specify that this is the final time point concentrations.

Line 104: I don’t know what these percentages in parentheses are referring to – I’m guessing it’s 1.99 times the control? I think this could be a bit more precise.

Line 106: Specify that the +Fe was higher than the +FeMn.

Line 109-111: This is misleading, since your statement should really be comparing +Fe to +FeMn,

which from your data looks like there is no difference.

Line 120: Refer to the actual changes in growth rather than percentages here, from looking at your figure it doesn't look like a huge change in Frag growth.

Line 134-136: The effect size of the MnFe compared to Fe alone is very modest.

Paragraph 137: no reference to figure.

Line 141-142: refer to the actual growth rates, the effect size looks quite modest.

Line 155: The data from this paragraph should be presented as a figure I think. For example, the difference between P in +Fe to +FeMn has a very small effect size, which would be evident from a figure.

Line 173: What does these lines represent? If you're going to fit a linear model, as these lines imply, that is a separate analysis than a correlation as you've described.

Line 184: I do not agree that you have provided evidence Mn is a "major driver of Southern Ocean phytoplankton distribution". You didn't really look at phytoplankton distributions, so I'm not sure how you can claim that. Also, the statement that picoeukaryotes at station east and Frag at station west were "clearly colimited" I think is quite a stretch. Overall the additional benefit to inferred growth rate was quite patchy when comparing +Fe to +FeMn.

Line 188: There needs to be a discussion about the disagreements between your two incubation experiments.

Line 230: While I do agree, your language throughout on the effect of MnFe colimitation is too strong.

Line 251: This is a bit vague. What about downregulation of Chl-a per cell? To me that seems like the most obvious reason.

Line 289: 'based on Moore et al 2013'... is that really the takeaway? I could be misremembering, but the 1:1 ratios of dFe and dMn in Moore et al 2013 were evidence of which elements in the ocean tend to be limiting. In any case, it's not clear to me what a 1:1 ratio of dFe:dMn is optimizing? (Since you've used the word 'optimal'!)

Line 321: The language is way too strong for a correlation of 11 datapoints.

Line 43: might be helpful to specify MnSOD?

Line 360: GEOTRACES all caps

Line 386: "physiological health" is vague, maybe specify photophysiological or be more precise?

Line 415: do these numbers represent standard error or standard deviation?

Line 456: I believe there is a mistake in this equation? I don't think there are two natural logarithms, only one. I think the equation should be:

$$\mu = (\ln(N_2 - N_1)) / dt$$

Line 491: specify the response variable.

Line 499: equation 4 is mentioned but is not present anywhere in the text.

Dear reviewers,

We have responded to all of the raised comments and have revised the manuscript:” *Phytoplankton species composition is governed by both iron and manganese limitation in the Drake Passage*”. We would like to thank you for the helpful comments, which greatly increased the quality of the paper and helped to lay out our findings more clearly. Our responses are below and marked in **purple**. To refer to a specific comment raised by a reviewer, the reviewer’s comment together with our response is denoted in the following: **R1_1** (reviewer 1 comment 1) and so on. We have highlighted all changes in **grey** colour in the denoted file “Article”.

Response to reviewers' comments:

Reviewer #1 (Remarks to the Author):

This paper tackles the question of the role of iron and manganese in shaping phytoplankton activity and community composition from a set of experiments conducted in Drake Passage. The results suggest evidence of Fe-Mn colimitation and an inhibitory effect of Mn addition on Phaeocystis. These results emerge from a set of standard bioassay experiments at two locations, for which phytoplankton community composition and photophysiology (Fv/Fm, sigmaPSII) were measured. Sampling at a set of additional stations revealed no link between dFe and Fv/Fm, while there was a positive correlation between Fv/Fm and dMn, and a negative correlation with dFe:dMn.

We thank reviewer 1 for the comments, which we have addressed in detail below.

R1_1: Right now, I do not think the paper is ready for publication in Communications Biology. It reads too much like a report, without sufficient mechanistic insight for a broad audience. There is a lot of description of the result associated with different phytoplankton groups and a set of photophysiological measurements, but we are never provided with much detail on mechanisms and processes. The new understanding of how the Southern Ocean system operates is, right now, not there. I was hoping for this to emerge in the discussion section, but this ended up largely being a restating of the results. → We agree with the reviewer that the discussion was too much reading like a restatement of the results. To address the reviewer’s point, we have modified the discussion and have completely re-written the discussion parts 2 and 3 (From line 212 to 315).

R1_2: For instance, I do not think it is new knowledge to state that the Southern Ocean is a region with high potential for Fe-Mn colimitation. As the authors know, this was first proposed by Martin decades ago. → We agree with the reviewer that Martin already hypothesized that the Southern Ocean is a region with high potential for FeMn co-limitation in the early 90ies (line 61), but it was only recently that clear evidence for Fe-Mn colimitation of phytoplankton growth in the Drake Passage was provided by Browning et al. (2021). In the latter study, however, no concomitant effect of Fe and Mn limitation on SO phytoplankton species composition was found, potentially due to the short incubation time between 2 up to maximal 5 days. Hence, clear evidence how Mn availability influences SO phytoplankton growth at the species level and thereby shapes SO phytoplankton composition is still lacking in the field and our study is the first to show that next to Fe Mn is an additional driver of SO phytoplankton composition.

R1_3: The link between Fe/Mn additions and the response of different phytoplankton groups and bacteria is interesting, but it is again a descriptive part of the paper with limited mechanistic insight. For example, the authors state that the low response from bacteria is because organic carbon is limiting, but was that measured during the experiments? → Please note that during the same expedition at the same two stations and from the same depth (West and East) Blanco-Ameijeiras et al. (2020) performed Fe-ligand experiments with the same phytoplankton communities revealing that the enrichment with exopolymeric substances, alternate carbon sources, increased the abundances of heterotrophic bacteria in addition to an enhancement of POC production. Since no Fe limitation effect was found, this points towards a possible carbon limitation of both communities (East and West). The latter

finding was also supported by several other studies on heterotrophic bacteria of the SO (eg. Church et al. 2000, Obernosterer et al. 2015, Fourquez et al. 2015, Fourquez et al. 2020). To make this point clearer in the revised manuscript, we have now written line 216-221: “Indeed, during the same expedition, Blanco-Ameijeiras et al. (2020)³⁵ performed Fe-ligand addition experiments with the same two natural plankton assemblages at station West and East. Their study showed that exopolymeric substances additions stimulated growth of heterotrophic bacteria at location West and East, most likely alleviating dissolved organic carbon limitation. Similarly, other studies have also shown that while low Fe concentration had no impact on heterotrophic bacterial communities^{36,37}, carbon and not TMs, may limit their growth in HNLC regions^{38,39}.”.

R1_4: Equally, why do the authors think different groups are responding differently to additions of Fe, Mn and both Fe and Mn? There is some (very brief) discussion of surface area to volume, but otherwise I was left unclear as to what was driving the results? This part of the paper should be strengthened significantly. → Certainly, we agree with the reviewer this part needed to be strengthened and therefore we have completely revised part 2 of the discussion (from line 212 to 275). Different to the submitted version, in the revised manuscript we now discuss why the diatom *Fragilariopsis* sp. responded differently at the two locations to the Fe/Mn additions and the brief discussion of surface area to volume has been removed. It now reads line 232-243: “...*Fragilariopsis* sp., which dominated the microphytoplankton community, reached its highest growth only when Fe was added in conjunction with Mn (Fig. 3c). This suggests a relief of FeMn co-limitation of *Fragilariopsis* sp. at West. No FeMn co-limitation on growth of *Fragilariopsis* sp. was observed at East. At this location, *Fragilariopsis* was possibly biochemically co-limited by Fe and other nutrients (Type III²⁷). In fact, the combined addition of FeB₁₂ to the same phytoplankton community at station East resulted in significantly higher growth of *Fragilariopsis* sp. (~0.12 d⁻¹) when Fe was added alone (Koch et al., unpublished data). This explains why the addition of FeMn in our study could not satisfy the vitamin B₁₂ requirement of *Fragilariopsis* at station East. Hence, we suggest that at both locations an alleviation from Fe limitation of *Fragilariopsis* might increase its requirement for other TMs like Mn in the case for station West or vitamin B₁₂ at station East, promoting maximum growth only when Fe was added together with the co-limiting nutrient.”

Moreover, we have now discussed in more detail what could be the reason for the observed growth inhibition *P. antarctica* at station West, please see line 249-263 in the revised manuscript.

R1_5: Lastly, the links between photophysiological measurements like F_v/F_m and some bulk micronutrient measurements are also not elaborated. This is an intriguing results, but the discussion is also underwhelming as it spends too much time restating results. I was left confused as to why dMn should correlate with F_v/F_m and I question whether this is at all causal. I was hoping / expecting that this part could be expanded with a wider Southern Ocean dataset. → The discussion of dTMs and F_v/F_m was not elaborated enough due to a small amount of data and a lack of mechanistic understanding. To account for the reviewer’s point, we extended the dataset with 11 additional *in situ* stations from another expedition (Polarstern expedition PS112) where we also sampled dFe, dMn and F_v/F_m (please see Table 4 and Fig.5). We have thus modified and re-written this part of the discussion (line 291 to 315): “To assess whether low F_v/F_m values reflect a limitation by multiple TMs, especially Fe and Mn, we sampled 22 contrasting *in situ* stations across the Drake Passage and along the Western Antarctic Peninsula (Fig. 1; Table 1,4). Across all sites, a wide range of dFe (0.05 to 4.51 nM) and dMn (0.08 to 6.92 nM) concentrations as well as of F_v/F_m values (0.12 - 0.56) was found (Table 3,4). Considering that F_v/F_m is commonly used as a proxy for Fe limitation in the field^{29,52}, it can be assumed that low dFe concentrations should be correlated with low F_v/F_m values, something which was, however, not the case across the 22 sampling stations (Fig.5a). As the sampling occurred in late summer/early autumn, and coincided with the time in which periodic dust inputs from Patagonia have already been reported⁵³, this lack of correlation could be potentially explained by these sporadic events. Iron associated to dust can be poorly soluble and not very bioavailable to SO phytoplankton⁵⁴, leading thus on the one hand to relatively high dFe concentrations, but on the other hand to low F_v/F_m values. Mn is required in the oxygen-evolving complex¹⁷, and Mn-limited diatoms have previously been reported to have compromised photosynthetic efficiency of PSII, resulting in low F_v/F_m values⁴⁸. In fact, there was a significant positive correlation between F_v/F_m and dMn concentrations across all stations (Fig.5b). Furthermore, our results indicate that Mn availability also influences the photophysiological signature of Fe-limited phytoplankton in the field. Fe limitation can cause higher oxidative stress and thus increase the Mn-

demand of phytoplankton⁴⁸. Therefore, the degree of Fe limitation phytoplankton experienced in the Drake Passage could be enhanced due to the low dMn concentration observed here (0.05 to 1.21 nM). Together the results from our observational and experimental data, highlight that F_v/F_m might not only be a proxy for Fe limitation, but may also be a photophysiological signature for a TM co-limitation with Fe, in our case for Fe and Mn. Hence, TM co-limitation by phytoplankton can be difficult to dismantle in the field purely on the basis of low F_v/F_m values alone and can only be confirmed through the performance of in situ TM addition incubation experiments. Clearly, further investigations are needed in order to assess the occurrence of co-limitation by Mn and other trace nutrients together with Fe in the SO.”

Figure 5. Linear regression (red line) between the dark-adapted maximum photosystem II quantum yield (F_v/F_m) and (a) the concentrations of total dissolved iron (dFe) or (b) the concentrations of total dissolved manganese (dMn). The data were determined at 22 in situ stations sampled across the Drake Passage and around the Western Antarctic Peninsula, including BIO1 as station West and BIO3 as station East. The two experimental stations West and East are indicated by dark circles while the PS97 stations are denoted by white circles and the PS112 stations by white triangles.

Reviewer #2 (Remarks to the Author):

Brief Summary of the manuscript: Balaguer and co-authors present evidence for phytoplankton species composition control by both iron (Fe) and manganese (Mn) in the Southern Ocean. Results come from shipboard bottle incubation experiments conducted in the Drake Passage where Fe and Mn were added alone and in combination. Authors further show that specific phytoplankton groups found in the Drake Passage such as the diatom *Fragilariopsis* were Fe-Mn co-limited, while other groups were only Fe-limited. Ultimately, these results support the availability of multiple trace metals being a major control for phytoplankton community growth and structure and not only Fe. These are some of the first natural (whole) phytoplankton community experimental results that show that the relief of Fe-Mn co-limitation in some parts of the Southern Ocean, recently shown by Browning et al. (2021), not only promotes total community growth and biomass increase but also responses from specific phytoplankton groups/species.

Overall impression of the work: I think the results presented in this manuscript are very interesting and an important contribution to the current state of knowledge regarding the role of low trace metal concentrations in the Southern Ocean. This in combination with the evidence from Browning et al., 2021 will likely bring about many new studies and objectives for research cruises and subsequent evaluation of trace metal parameters in current ocean biogeochemical models. I applaud the combined use of observational results from other in situ stations with that from the two experiments to provide better context.

However, some parts of the manuscript read difficult and do not flow smoothly. For example, a few abbreviations are not defined when first used. Although not essential, authors should consider reducing the use of abbreviations that are not used often. The visual presentation of results could be improved with both edits to existing figures (e.g. adding colour to bar plots). Additions to the method section is also suggested to both improve readability

and ensure common queries/concerns of bottle incubation experiments are addressed. Also, due to the structure/format of this journal (methods at the end), a short description of each variable could be useful when first discussed in results to improve readability/context and better inform the reader. There are a few statements in the discussion that are not properly explained and/or unclear. Specifically, I suggest to further address in the discussion the likely reasons that could have led to experiment results indicating that *Phaeocystis* was inhibited by Mn. Brief mentions were already made to the uncertainty for why *Phaeocystis* appears to be inhibited. If this is the case, I would suggest rephrasing Line 20 in the abstract to be less direct. Below are specific comments and suggestions.

We thank Mr. Viljoen for the all the positive and detailed comments. As all the comments raised here were again specifically mentioned in more detail below, you will find our responses below, which specifically address them. Only with respect to his comment why growth of *Phaeocystis* was inhibited, we have modified the respective part of the discussion and it now reads in line 249 to 263: “*While the single addition of Fe significantly enhanced growth of the bloom-forming species P. antarctica at West, the combined addition of FeMn did surprisingly not trigger its growth (Fig. 3c). Similarly, growth of P. antarctica was inhibited when Fe was added in combination with Zn and Co to the same phytoplankton community (Koch et al., unpublished data). Only when Fe was added together with vitamin B₁₂, maximum growth rate of P. antarctica was achieved at West, indicating relief from FeB₁₂ co-limitation (Koch et al., unpublished data). While we are unsure how to explain these effects, top-down and bottom-up controls such as growth inhibition and/or competition could be an explanation. The very low dFe concentrations at West (0.05 nM) potentially triggered competition for Fe among the different members of the community⁵⁰. As the bloom-forming diatoms are generally considered as r strategists²⁵, they usually have high resource uptake capabilities, thus providing them with a competitive advantage over other groups. Due to its much higher cell abundance relative to P. antarctica at West (Suppl. Table 2), the bloom-forming Fragilariopsis sp., may have been more efficient than the prymnesiophyte in taking up the added Fe and Mn, potentially causing the observed diminished growth rates by P. antarctica (Fig. 3c). Similar to our findings here, during a diatom-dominated bloom, strong competition for dFe between microbes, small phytoplankton and diatoms was reported⁵⁰.*” Please note that the line 20 have been removed from the abstract as it is not the main message that we would like to share there.

Introduction:

R2_1: Line 30: Rephrase to “The Southern Ocean is unique compared to much of the global ocean as it accounts for over 40% uptake of anthropogenic carbon dioxide (CO₂), making it the largest oceanic CO₂ sink (1,2)” If space allows, add one sentence on what role phytoplankton plays in this uptake and add Buesseler et al. (2020) as a reference who recently reported that the role of phytoplankton and quantity of carbon sequestration occurring in the Southern Ocean is underestimated. → As suggested we have modified this sentence and have also included the studies by Dunne et al. and Buesseler et al., it now reads in line 30-33: “*The Southern Ocean (SO) is unique compared to much of the global ocean as it accounts for over 40% uptake of anthropogenically derived carbon dioxide (CO₂), making it the largest oceanic CO₂ sink^{1,2}. With 10% being biologically fixed³, the amount of biological sequestered CO₂ by SO phytoplankton is still underestimated⁴.*”

³ Dunne, J. P., Sarmiento, J. L., and Gnanadesikan, A. (2007). A synthesis of global particle export from the surface ocean and cycling through the ocean interior and on the seafloor. *Glob. Biogeochem. Cycles* 21:GB4006.

⁴ Buesseler, K. O., Boyd, P. W., Black, E. E., & Siegel, D. A. (2020). Metrics that matter for assessing the ocean biological carbon pump. *Proceedings of the National Academy of Sciences*, 117(18), 9679-9687.

R2_2: Line 31: Note first use of “CO₂”. Please define in text here. → As suggested, the abbreviation was added in line 31.

R2_3: Line 32: Define “SO” abbreviation earlier in Line 30. Also rephrase to “...SO (3), which results in the SO being characterized as the world’s largest...” → As suggested by the reviewer, the abbreviation has been introduced in line 30 and the sentence in question has been rephrased in line 34-35 to: “... *primary production remains low in the SO⁵, making the SO the world’s largest High-Nutrient Low-Chlorophyll (HNLC) region.*”

R2_4: Line 33: Suggest adding short sentence before “Low essential trace metals...” about trace metals as important micronutrients to phytoplankton with a reference such as Twining & Baines (2013). Currently the use of “essential” is unclear. Also suggest to rephrase to: “Low concentrations of essential trace metals (such as iron,...”. → We agree and we have added this wording and the suggested references in line 35: *“In addition to macronutrients, trace metals are also needed for several physiological processes and therefore influence phytoplankton growth⁶.”*. The sentence starting with “Low concentrations...” has been removed to avoid repetition with line 48.

R2_5: Line 36: Change “biomass build up” to “growth”. → This has been changed in line 39.

R2_6: Line 37: Change “different” to “various” → This has been changed in line 40.

R2_7: Line 41: Change “trace metal needed in the...” to “trace metal needed during photosynthesis in the...” → As suggested, this was changed in line 44.

R2_8: Line 43: Expand on why the scavenging of ROS is important. → Indeed this was unclear, accordingly the sentence has been modified in line 47 to: *“...which scavenges reactive oxygen species during photosynthesis, thus preventing potential cell damage¹⁷.”*

R2_9: Line 45: Change “Just like Fe,” to “Similar to Fe,” → This has been changed in line 47.

R2_10: Line 47-48: Be specific on what type of studies is referred to here: observational or experimental. Maybe rephrase to “Several studies have subsequently suggested the occurrence of Mn co-limitation with Fe of SO phytoplankton”. → We agree, indeed it was unclear, this has been changed in line 50 and it now reads: *“Several trace metal (TM) bottle enrichment experiments in the SO have indicated the occurrence of a Mn co-limitation with Fe of phytoplankton^{7,19,21-23}.”*

R2_11: Line 55: Not sure if it is necessary to point out here in the introduction what form of Fe “FeCl₃” was used in additions. → We agree with this suggestion and FeCl₃ has been removed in line 58.

R2_12: Line 58: Add “the” before “Drake Passage” → This has been corrected in line 61.

R2_13: Line 58: Is there a reference on the time it takes for a community change after a change in their environment? → Please note that Browning et al. 2014 mainly provided data of Fv/Fm and chlorophyll *a* biomass and did not investigate species composition.

R2_14: Line 67: Change “shapes” to singular “shape” → As “*shapes*” refers to “*Mn availability*” in line 72, we would like to keep this sentence as it is.

R2_15: Line 73: Change to “... and picoeukaryotes in natural communities sampled at two different locations...” → We changed to whole sentence in line 78-80 to: *“Our results show that in two natural communities of the Drake Passage only the growth of some diatoms (*Fragilariopsis* sp.) and picoeukaryotes were limited by both Fe and Mn while the other phytoplankton groups were primarily limited by Fe.”*

Since the results from this study might also lead to discussions about the single and/or co-limitation of other metals as well, it is worth briefly mentioning previous work that discussed/suggested co-limitation of multiple metals other than Fe if length restrictions allow. → This is an interesting point indeed, which was included in line 62 as an introduction for different types of co-limitation: *“Due to different TM availability in the seawater and different TMs requirements by the cells, phytoplankton can be limited by multiple TMs^{6,24} thus leading to co-limitation by different TMs at the community level²⁵.”*

Results:

R2_16: Line 80: Make sure the abbreviations “Chla, dFe and dMn” are defined at first use. For example: “dissolved iron (dFe)”. Also review units of Chla in Table 1. Chla units in mg/L. $0.09\text{mg/L} = 90\text{ug/L}$. This seems unlikely. Please explain or revise. → As suggested by the reviewer, the abbreviations have been introduced in line 85-86. Moreover, the reviewer is right, there was indeed a unit mistake in Table 1, which has been corrected and is now ug/L.

R2_17: Line 93: be more specific here and state that picoeukaryotes were determined via flow cytometry. At this stage of the manuscript, it is unclear why picoeukaryotes are defined and grouped separately from the microphytoplankton. Perhaps state briefly when first discussed how each was determined. Also, define here the differences between the various P groups. → Indeed, it is unclear how we defined picoeukaryotes. To clarify this, it now reads in line 98-99: “*For both experiments, picoeukaryotes (P), nanoeukaryotes (N) and heterotrophic bacteria (B) were determined via flow cytometry, with the three subgroups of P defined according to their sizes (P1, P2 and P3) (Suppl. Fig.1).*”

R2_18: Line 95 to 96: Again, briefly mention how N and B were determined. Perhaps revise the first sentence of the paragraph at Line 93 to state that Flow cytometry analysis revealed that P, N and B were present at both locations. → As suggested by the reviewer, this has been changed, please see our response given to comment **R2_17**.

R2_19: Line 101: “nitrite/nitrate” Is this the sum of nitrate and nitrite or just nitrate. See the method section for further comments. In tables, NO_x is defined but not in text as far as I can see. At first use of Nitrate in results define that it refers to the sum of nitrite and nitrate. → Indeed, it was unclear and accordingly it was now defined in line 84-85: “*high concentrations of macronutrients ($\text{NO}_x = \text{nitrate } (\text{NO}_3) + \text{nitrite } (\text{NO}_2)$); phosphate (PO_4); silicate (SiOH_4)*”.

R2_20: Line 104: Define here how chla accumulation rates are calculated. Especially since detailed methods are at end for this journal structure. It might provide better context to the non-expert reader if similar figures to Fig2 and Fig3 was added to the supplemental showing the absolute chla concentrations and cell abundances at the beginning and end of each treatment and control for both experiments. → To account for the reviewer’s point, it now reads in line 108: “*Chla-based accumulation rates, growth rates of three P groups derived from flow cytometry as well as of the different microphytoplankton genera based on light microscopy were calculated using Eq.1 (cf. Methods section)*”. As asked by the reviewer, absolute cell abundances determined via flow cytometry and microscopy as well as absolute chlorophyll *a* concentrations are now shown in the Supplementary Material, please see STable 1, STable 2 and SFig. 2.

R2_21: Line 112: Refer to earlier comment relating to defining the P groups. → This has been done, please see our response given to comment **R2_17**.

R2_22: The previous suggestion to add figures showing the absolute abundances of each method would especially be useful to gain a better perspective on the negative growth rates shown. → This has been done, please see our response given to comment **R2_20** (STable 1, STable 2 and SFig. 2).

R2_23: Again, briefly define/explain here how growth rate is determined with flow cytometry results or provide a detailed explanation of this in the methods section and refer to the specific method section here. This applies to all growth/accumulation rates used. → As suggested by the reviewer, this has been done, please see our response given to comment **R2_20**.

R2_24: Line 145: Visually it seems chaetoceros growth decreased at +Mn treatment compared to control? Please confirm. → The reviewer is right that growth of *Chaetoceros* sp. decreased in the +Mn treatment compared to the Control. This was not clearly stated in the results section. To point out this finding, it is now written in line 149: “*The growth rate of Chaetoceros sp. decreased after the addition of Mn, ...*”).

R2_25: Line 155: Depending on the length, I would suggest keeping the same structure as previous sections: “Experiment West”, “Experiment East” and then the additional section here “in situ stations”. → We agree with the reviewer and now point out when we refer to “Experiment West”, “Experiment East” and the “*In situ* stations”.

R2_26: Line 158 and photophysiological results in general: Reporting summary states "The physiological health of the community was assessed every 2-4 days and enable to determine the day of the harvesting (=sampling)." However, this is not clearly stated in the methods of Fv/Fm (Chlorophyll a fluorescence). If this is the case, an additional figure of Fv/Fm and SigmaPSII over time in the supplemental would be beneficial. This could be used to briefly discuss photophysiological status overtime after additions that due to the length of the experiment the Fv/Fm at the end of the treatment might have changed during the experiment. Viljoen et al. (2018) showed a rapid increase in Fv/Fm within one day after Fe addition treatment of Antarctic waters whereafter Fv/Fm reached a plateau after 2-3days and then began to decrease. → Please note that in the submitted manuscript, it was wrongly stated that final sampling was determined on the basis of changes in Fv/Fm. “*As in previous Fe addition experiments, final sampling took place between 7 and 15 days depending on the treatment, with Fe enriched treatment harvested first*^{53,62,63}.”, this is now written line 386. To account for the reviewer’s point nevertheless, we added a figure in the Supplementary Material showing the development of Fv/Fm over time for both experiments (Supplementary Fig. 3).

Discussion:

R2_27: Line 192: Change “macronutrients” to singular “macronutrient” → This was corrected in line 199.

R2_28: I like Table 1 as it provides a good overview of initial conditions. However, authors should consider adding the initial Fv/Fm values for each experiment from Table 4 to Table 1 as well. → To account for the reviewer’s suggestion, we now show, in addition to table 4 (now table 3), the initial Fv/Fm values for each experiment in Figure 1 as well. Here, for both stations initial concentrations of dissolved iron (dFe), dissolved manganese (dMn), chlorophyll a (Chl α) as well as the 1h-dark-adapted maximum photosystem II quantum yield (Fv/Fm) are now also shown.

R2_29: Line 193-195: Sentence is a bit unclear. Were dMn concentrations of this study lower than what was measured by others for the Drake Passage? Or was dMn below 0.2nM, similar to previous studies. → The sentence indeed led to misunderstanding, therefore we changed the sentence in line 200: “*Concentrations of dMn at our two sampling sites (Table 1) were low, being below 0.20 nM, and similar to previously measured values across the Drake Passage*^{18,20,21,23}.”

R2_30: Line 196: Change “indicative for” to “indicative of”. Change “In line with this,” to “Accordingly,” → This was corrected in line 202 and 203, respectively.

R2_31: Line 200: Change “predominant” to “dominant” → This was done in line 206.

R2_32: Line 203: Insert “low” before “biomass, plankton abundances and ...” → As suggested, we have modified the sentence in question in line 209: “*Biomass, plankton abundances and dTM concentrations were all low, suggesting...*”.

R2_33: Line 214-215: A reference to the earlier suggested supplemental absolute abundance figures (Fluorometer Chl α , Flow cytometry and microscope determined micropphytoplankton) would be useful here to provide better context for the total abundance reached for the Fe only addition treatment and others. → This was done, please see our response to comment **R2_20**.

R2_34: Line 222-223: Rephrase sentence starting “Since the addition...” to “The addition of Fe led to similar responses as the addition of both trace metals together, which indicates that Fe...” → As part 2 of the discussion was changed completely, this sentence does not exist anymore.

R2_35: Line 224-226: Statement a bit unclear. It is unclear to me how the stimulation in growth of the two P groups due to Fe addition show how small cells better cope with low Fe. I suggest rephrasing in some way to improve clarity. → As part 2 of the discussion was changed completely, this sentence does not exist anymore.

R2_36: Line 230-231: Rephrase sentence starting “This result clearly shows...” to “Therefore, the diatom *Fragilariopsis* sp. was clearly co-limited by both Fe and Mn → As part 2 of the discussion was changed completely, this sentence does not exist anymore.

R2_37: Line 237: Add “an increase in” after “stimulated” → As part 2 of the discussion was changed completely, this sentence does not exist anymore.

R2_38: Line 239: Change “...and was attributed to...” to “was mostly attributed to” → As part 2 of the discussion was changed completely, this sentence does not exist anymore.

R2_39: Line 241: This is an interesting point. I might have missed it in a later section, but I suggest expanding here or in the dFe:dMn ratio section later on the possible reasons for why P2 is co-limited at East where dMn is much lower than West where *Fragilariopsis* was only species/group co-limited. → This is a very interesting point and the reviewer is right that “*the very low dMn concentration observed at station East (0.08 nM) possibly enhanced the Fe demand for growth of this group, which was only relieved when Fe and Mn were amended together*” (line 269). Please note that the reasoning why P2 was co-limited at both locations is now discussed more thoroughly in the revised manuscript. This section now reads in line 264-275: “*Different to location West, the analysis of the bulk parameter Chla indicated a Fe-Mn co-limitation effect at the community level. In this case, Chla-based accumulation rates of all and small cells (< 2 μm) reached highest values only when both Fe and Mn were added (Fig.4a). The same trend was also observed for the picoeukaryote group P2, which numerically dominated the East community (Fig.4b) and thus contributed most to the Chla pool (Fig.4a). These results indicate a biochemical dependent co-limitation between Fe and Mn (Type III²⁷) of the P2 group at East. The very low dMn concentration observed at station East (0.08 nM) possibly enhanced the Fe demand for growth of this group, which was only relieved when Fe and Mn were amended together. Thus, due to different cellular Fe and Mn requirements and TMs availability⁶ not all members of a community may be affected equally by co-limitation (also defined as community co-limitation²⁵). As a consequence, this can result in regional and interspecific differences in cellular phytoplankton stoichiometries and possibly lead also to a cascade effect of limitation by trace metals other than Fe and Mn^{6,24}.*”

R2_40: Line 252: Explain what is meant by “competition”. Are the authors referring to the competition between species/groups or metal competition? Expand on why the decrease in *Phaeocystis* growth is more pronounced in East experiment than West. Include possibilities of pre-acclimation to certain initial conditions that might have led to this result. If related to the dFe:dMn ratios discussed later, refer the reader to that section. Refer to a combination of metal availabilities at these stations i.e. what species have absolute requirements vs can substitute Mn for another metal. This can be introduced in introduction by referring to Twining & Baines (2013), observational study Viljoen et al. (2019) and experimental study of Browning et al (2014) for the possibility of multiple nutrient limitations. Also, some studies already referenced here show the importance of Mn as an absolute requirement for most phytoplankton. This could possibly be used here as support for Mn not inhibiting *Phaeocystis* growth, but rather the competition by dominant diatoms for finite nutrient resources in bottle incubation experiments. Are there initial or in situ concentrations for other metals available? Or nutrient concentrations at the end of experiments to see possibility of other limiting macro- and micronutrients? → We refer to “...*competition for Fe among the different members of the community*” in line 256. Please note that this part of the discussion has been completely rewritten to discuss more thoroughly why growth of *P.antartica* declined in line 249-263. As suggested by the reviewer we now also included information on the effect of additions from other trace metals than Fe and Mn, such as Zn, Co and vitamin B₁₂ on growth of *Phaeocystis*, all tested for the same two communities by Koch et al. (unpublished data). It now reads in the revised manuscript in line 250-254: “*Similarly, growth of P. antarctica was inhibited when Fe was added in combination with Zn and Co to the same phytoplankton community (Koch et al., unpublished data). Only when Fe was added together with vitamin B₁₂, maximum growth rate of P. antarctica was achieved at West, indicating relief from FeB₁₂ co-limitation (Koch et*

al., unpublished data).” Moreover, we have also included data on the initial concentrations of Zn and Co as well as of the dFe:dZn and dFe:dMn of each station in Table 1. Furthermore, the final macronutrient concentrations are available in the supplement (please see Suppl. Table 3), but there was no significant drawdown relative to the initial values (Table 1).

R2_41: Line 253: Reference Twining & Baines 2013 at end of sentence. → As suggested, the reference was added in line 275.

R2_42: Line 255: Fv/Fm discussion section: Currently the connection between Fe availability and Fv/Fm needs to be expanded. First sentence is a nice overall intro to this section. However, after this a short example of how Fe impacts Fv/Fm and why would link the first sentence to the rest of the paragraph better. Or expand first sentence to include (remind reader) how Fv/Fm fits into this statement. One suggestion: change Line 257 to "... transport chain (14), and typically results in decreased Fv/Fm and increased PSII (strzepek 2012; Behrenfeld and miligan, 2013)" → We agree, this has been changed as recommended in line 277: *“Reduced Fe availability usually impacts photosynthesis, as it is essential in many parts of the electron transport chain¹⁶, typically resulting in decreased F_v/F_m and increased σ_{PSII} of phytoplankton cells^{13,51}.”*

R2_43: Line 258-259: Comment on why West shows a significant difference in both Fv/Fm and SigmaPSII and East only Fv/Fm in Table 4? Author’s state East likely more Mn co limited than West. Could it be due to higher percentage Fragilariopsis diatoms in West than in East? → Unfortunately, we cannot provide any explanation why location West showed a significant difference in both Fv/Fm and SigmaPSII while at location East only Fv/Fm responded to Fe and Mn addition. As the photophysiological signature (Fv/Fm and SigmaPSII) reflects the response of all members of a phytoplankton community, hence at the community level, we believe it is not possible to identify what was driving this photophysiological response.

R2_44: Line 277: Sentence started with symbol/abbreviation “Fe”. Rather start sentence with “Iron”. → This was changed in line 299.

R2_45: Line 289: Add “to be” after “expected” → As part 3 of the discussion was changed completely, this sentence does not exist anymore.

R2_46: Line 292: Rephrase to For example: "However, due to the wide range of dFe:dMn ratios observed for both coastal and open stations (Fig. 4c), one cannot attribute Mn co-limitation to specific marine areas for this study." Add mention of how this observation/statement compares to what Browning observed for coastal vs open. Is this in contradiction or support their observations? Perhaps use this to further support the next sentence. → Please note that this part of the discussion was completely revised in line 291-315. Moreover, the part on dFe:dMn ratios was removed as it did not provide any additional information.

R2_47: Line 297: Mention here or in introduction what the other less major drivers could be for the occurrence of HNLC zones. → Indeed, light and grazing also influence the productivity of HNLC regions. To account for this, the sentence was modified in line 317-329: *“Although light and grazing by microzooplankton are important factors, since 1990¹⁰ it has been demonstrated that Fe is the predominant driver controlling phytoplankton productivity in HNLC regions⁵⁴. Recent studies highlighted that in addition to Fe other micro-nutrients such as Mn, cobalt, zinc and vitamin B₁₂ can also influence phytoplankton biomass^{21–23,28,47,55}.”*

R2_48: Line 309: Change “metals” to singular “metal”. → This was corrected in line 329.

R2_49: Line 315: It was suggested in text earlier that there are other reasons why P. antarctica could have appeared to be inhibited in the Mn only experiment. Refer to previous comments. Studies have shown that phaeocystis do have a requirement for Mn next to Fe such as for ROS enzyme (Twining & Baines 2013). Maybe expand on possibility for other metal/resource competition, towards the end of such a long experiment. → Please see our answer given to comment **R2_40**. Moreover, potential co-limitation effects by other nutrients were

included, please see the rewritten part 2 of this discussion. In addition, please note that this sentence has been removed on the new manuscript line 331.

R2_50: Line 332: End current sentence after “structure” here and start another sentence “Therefore, dFe must not be the only TM considered...” → This was corrected in line 350 (“...community structure. Therefore, dFe must not be the only TM considered when...”).

Methods:

R2_51: Line 354: Mention why sampling depth was set to be 25m. Reduce contamination possibility from ship in shallow waters? Logistical reasons? → At both sites Indeed, reduced contamination was the main reason. This was now added line 371.

R2_52: Line 377: Many other studies did dFe additions of 1nM or more (e.g. Viljoen et al. 2018). Browning et al. 2021 even had 2nM for both dFe and dMn. Please explain and justify why these concentrations were chosen with reference to other studies. → The choice of such low Fe addition was made in order to avoid precipitation of Fe. To clarify this, it is now written in line 384: “Low dFe and dMn additions were made in order to avoid formation of inorganic colloids in our experiments. Fe chemical speciation data from a joint study⁶⁰ during PS97 demonstrated that in-situ ligands were present in sufficient quantity to react with the FeCl₃ added.”

R2_53: Line 383: Add here whether bottles were mixed every few days or not and why. Was this maybe done at each FRRf measurement? → Yes, this has been clarified in line 395: “To avoid sedimentation of the cells, all incubation bottles were gently rotated every day.” Of course, before each sampling each incubation bottle was gently rotated.

R2_54: Line 384: Add details here on the light controlling equipment used and if known, the applicability of the emitted light spectrum for phytoplankton growth. → To clarify this, it now reads in line 393: “All incubation bottles were maintained at 30 μmol photons m⁻² s⁻¹ white light (providing the full light spectrum) under a 16:8 (light:dark) hour cycle...”

R2_55: Line 385: Add further detail on the variation in the length of each experiment/treatment and what the reasons were. → Please see our answer to **R2_26**.

R2_56: Line 405: Define “HNO₃” → As requested, the definition is given in line 410 (“double distilled nitric acid (HNO₃)”).

R2_57: Line 416-417: Define here that total nitrate (nitrite+nitrate) is used in results. Why is only silicate referred to as “dissolved silicate”? If macronutrient concentrations at the end of experiments are available, it could be useful to calculate nutrient drawdown rates during the experiment and compare to the various growth rates. → As suggested by the reviewer the definition was provided in line 421: “The dissolved macronutrient concentrations (total nitrate (nitrite+nitrate), phosphate and silicate) were determined ...”. Please note that the information on the concentrations of all macronutrients at the end of the experiment was already available in the Supplementary Material (Supp. Table 3) in the submitted version of this manuscript. There was, however, no significant macronutrient drawdown observed and all macronutrients remained largely in excess at the end of the experiments.

R2_58: Line 420: Add to Light microscopy and Flow cytometry method sections how growth rates were calculated for these methods/results. → Please see our response previously given to comment **R2_20**, where we specifically respond to this question.

R2_59: Line 443: If possible, define the size range for each P subgroup. → The size range for each P subgroup (P1-3) was determined according to the different chlorophyll (*a,b*) fluorescence, then from that contrasted sub-clusters of events were determined. The method is shown in the Supplementary Material Fig.1.

R2_60: Line 446: Add to the Chlorophyll-a method section the volume filtered for the fluorometer analysis. → To account for the reviewer's request, it is now written in line 458: "...on average 1000 ml at the beginning and on average 250 mL at the end of the experiments seawater were filtered...".

R2_61: Line 487: Mention specific R packages used and their reference (main contributors) or where specific code used can be found if uploaded to a repository. → Indeed, this information was missing and is now given in line 501: "R packages to reproduce statistical analysis are: stats (*aov*, *lm* and *cor.test* functions), ggplot2 (*ggplot* function) and agricolae (*HSD.test* function).

R2_62: Line 497: Give references to specific R packages used during statistical analysis to give recognition to its contributors and for readers to know what packages they can use. → Please see our response given to comment **R2_61**.

Figures & tables:

R2_63: Figure 1: A more informative and easier to read figure of the study area would be beneficial. For example, increase size or contrast of sample/experiment locations. Add data such as initial community Chla concentration for each location. Perhaps also add similar map figures as supplemental to visually show initial metal concentrations. Increase domain of map north and east. Include "West" and "East" notations on the map to visually remind the reader of how experiments are referred to in the text. → We completely agree with this suggestion and have changed Figure 1. Please see also our response given to comment **R2_28**.

Figure 1. 22 in situ stations were sampled across the Drake Passage and around the Western Antarctic Peninsula during the Polarstern expeditions PS97 (red dots), including BIO1 as station West and BIO3 as station East, as well as during PS112 (yellow dots). For station West and East, initial concentrations of dissolved iron (dFe), dissolved manganese (dMn), chlorophyll a (Chla) as well as the 1h-dark-adapted maximum photosystem II quantum yield (F_v/F_m) are shown in the dark box.

R2_64: Figures 2 & 3: Suggest to add colour to improve the contrast of error bars for readers to visually see the comparison between treatments. Error bars are not visible for the black bars (e.g. Total, P3 & Phaeocystis). → As suggested, **figure 2 and 3**, which are now **figure 3 and 4**, were modified to account for the reviewer's comment.

Figure 3. Community response at Station West. (a) Chlorophyll *a*-based accumulation rate (d^{-1}) of all (total), the small (0.2-2 μm) and the large (> 2 μm) cells, (b) net growth rate of each picoeukaryote group (subgroups P1-P3) determined via flow cytometry and (c) net growth rate of *Chaetoceros* sp., *Fragilariopsis* sp., *Pseudo-nitzschia* sp. and *Phaeocystis antarctica* determined via light microscopy. All parameters were estimated from the community sampled at station West (BIO1) after exposure to different Fe and Mn availabilities. Values represent the mean \pm SD. Different letters indicate significant differences between treatments ($p < 0.05$).

Figure 4. Community response at Station East. (a) Chlorophyll *a*-based accumulation rates (d^{-1}) of all (total), the small (0.2-2 μm) and the large (> 2 μm) cells, (b) net growth rate of each picoeukaryote group (three subgroups P1-P3) determined via flow cytometry and (c) net growth rate of *Chaetoceros* sp., *Fragilariopsis* sp., *Pseudo-nitzschia* sp. and *Phaeocystis antarctica* determined via light microscopy. All parameters were estimated at the end of the experiment from the community sampled at station East (BIO3) after exposure to different Fe and Mn availabilities. Values represent the mean \pm SD. Different letters indicate significant differences between treatments ($p < 0.05$).

R2_65: Table 2 could be turned into figure format to visually represent the contribution of each species/group for each location's initial community. For example, 4x pie charts (2 for microphytoplankton and 2 for picoeukaryotes) or in one barplot format. Essential to ensure that readers are properly aware of the initial community and/or can easily find it when just scanning through. The original table 2 could move to supplemental. → We agree that pie charts would be an easier way to visualize the starting communities. This had been done and the data are shown in **figure 2**.

Figure 2. Relative abundances of the four microphytoplankton genera (a, c) and of the three picophytoplankton subgroups (P1-P3; b, d) at the start of the experiment at Station West (a, b) and East (c, d) was determined via light microscopy (a, c) and flow cytometry (b, d).

R2_65: Table 1: As said in the main text comments, I suggest adding Fv/Fm values to Table 1. → Please see our response given to comment **R2_28**.

Reviewer #3 (Remarks to the Author):

General comments:

Balaguer and coauthors present results from two bottle incubation studies looking at the effects of manganese and iron additions in the Southern Ocean. To date, most bottle incubation studies conducted with iron and manganese only examine bulk community-level properties (e.g. total particulate carbon, total chl-a), while these authors present a more fine-grained analysis of taxonomic composition. While I think these results are interesting and the data are valuable, I have major reservations about the analyses conducted and the strength of the conclusions, and even the principal statement that there is a strong effect of Fe-Mn colimitation. There are also some issues with the statistical analysis and methods (see below) that should be addressed. However, I do think with major changes to the manuscript this will be a valuable contribution to the literature.

We thank the reviewer 3 for the valuable comments that helped to improve the manuscript. All comments by reviewer 3 are addressed below.

R3_1: My first reservation is conceptual. The authors state that Mn-Fe (together) is a “major control” on growth of SO phytoplankton. But they are only highlighting when effects of one experiment is significant, regardless if that outcome is mirrored in the second experiment. Presumably if something is a major controller of SO phytoplankton, there should be some replicability when two experiments are conducted in the same region of the

ocean? For example, to me the result that *Fragilariopsis* was Mn-Fe colimited in one experiment but not the other should indicate that Mn-Fe colimitation is actually not very common. I suggest limiting strong statements about conclusions only to those effects that were mirrored between the two experiments. → We agree that the use of words such as ‘major control’ were too strong and were removed throughout the manuscript. To account for the reviewer’s point, the whole part 2 of the discussion (from line 212 to 275) has been completely changed. Different to the submitted version, in the revised manuscript we now discuss why for instance the diatom *Fragilariopsis* sp responded differently at the two locations to the Fe/Mn additions.

R3_2: Another major reservation I have is with respect to the statistical analysis and data presentation (additional details below). There is a lot of emphasis on statistical significance, but the effect size is not emphasized. I would encourage the authors to state the effect size not in percent change, but rather the actual magnitudes of the effect in their appropriate units. I think this might significantly shift the main message of the paper, because the effect sizes comparing +Fe and +FeMn look quite small. Actually, when I first looked at the figures without reading the text I concluded the opposite that +Fe alone is the dominant effect, except in a small number of instances! → We agree that we mainly concentrated on statistical significances and not on the effect size in the submitted manuscript. As suggested by the reviewer, we removed the magnitude of changes that was expressed in % from the revised manuscript in the whole Results section. Moreover, to visually simplify the reading of the growth responses of each group we have modified the figures 3 and 4. Indeed, the reviewer is right that Fe was the main driver of the growth responses discussed in part 2 of the discussion, with only significant effects of FeMn together on the diatom genus *Fragilariopsis* at West and the picoeukarote group P2 at East. To make this point clearer, part 2 of the discussion has been changed accordingly, please see line 212-275. Now we point out more clearly that “*Fe was the main driver at the community level*” (line 230), but also that diatom genus *Fragilariopsis* at West and the picoeukaryote group P2 at East were co-limited by Fe and Mn.

R3_3: Lastly, I did not see a data availability statement. I think depositing the raw data in a public repository like Dryad should be done to maximize community benefit from these difficult-to-collect data! → Unfortunately, we forgot to mention that all data from this study were already submitted to the PANGAEA data base and will be freely available as soon as this manuscript has been published. This has been clarified in the Data availability section in line 506.

R3_4: This is not a reservation, but a thought. It’s interesting that in some cases the experiments had shifted community structure, but the bulk metrics look similar. I think this is an important contribution, since most bottle incubations looking at bulk properties would completely miss that! It might be interesting for the authors to expand on this point in their discussion. → We completely agree with the reviewer that while the bulk parameters (Chla) were similar it is very interesting to observe community changes. This latter aspect has been pointed out more clearly in the discussion in line 230-234: “*Even though for experiment West, the analysis of bulk parameters such as Chla indicated Fe availability as main driver, the microscopic analysis, however, revealed that the ecologically and biogeochemically important diatom Fragilariopsis sp., which dominated the microphytoplankton community, reached its highest growth only when Fe was added in conjunction with Mn (Fig. 3c).*” as well as in line 264-275: “*Different to location West, the analysis of the bulk parameter Chla indicated a Fe-Mn co-limitation effect at the community level. In this case, Chla-based accumulation rates of all and small cells (< 2 μm) reached highest values only when both Fe and Mn were added (Fig.4a). The same trend was also observed for the picoeukaryote group P2, which numerically dominated the East community (Fig.4b) and thus contributed most to the Chla pool (Fig.4a). These results indicate a biochemical dependent co-limitation between Fe and Mn (Type III²⁷) of the P2 group at East.*”

Overarching comments

R3_5: Figures should show raw data points, not bars +/- the mean. There is a lot of literature on the benefits of presenting data this way, and bar plots are not as transparent as showing the raw data. → To account for the reviewer’s point, next to the error bars, now each measured data point is also displayed in **figure 3 and 4**.

Figure 3. Community response at Station West. (a) Chlorophyll *a*-based accumulation rate (d^{-1}) of all (total), the small (0.2-2 μm) and the large (> 2 μm) cells, (b) net growth rate of each picoeukaryote group (subgroups P1-P3) determined via flow cytometry and (c) net growth rate of *Chaetoceros* sp., *Fragilariopsis* sp., *Pseudo-nitzschia* sp. and *Phaeocystis antarctica* determined via light microscopy. All parameters were estimated from the community sampled at station West (BIO1) after exposure to different Fe and Mn availabilities. Values represent the mean \pm SD. Different letters indicate significant differences between treatments ($p < 0.05$).

Figure 4. Community response at Station East. (a) Chlorophyll *a*-based accumulation rates (d^{-1}) of all (total), the small (0.2-2 μm) and the large (> 2 μm) cells, (b) net growth rate of each picoeukaryote group (three subgroups P1-P3) determined via flow cytometry and (c) net growth rate of *Chaetoceros* sp., *Fragilariopsis* sp., *Pseudo-nitzschia* sp. and *Phaeocystis antarctica* determined via light microscopy. All parameters were estimated at the end of the experiment from the community sampled at station East (BIO3) after exposure to different Fe and Mn availabilities. Values represent the mean \pm SD. Different letters indicate significant differences between treatments ($p < 0.05$).

R3_6: Figures showing statistical output of a Tukey's test should use the convention of a, b, etc. The many symbols are relatively confusing, and it's not clear with the *, # symbols. → We agree with the reviewer and now apply the convention of a,b,c in figure 3 and 4.

It's confusing to me why some taxa decreased in abundance from the initial time point. → In fact, growth of *Phaeocystis* was inhibited at station West. To discuss more thoroughly the reason for this finding, we have modified the respective part of the discussion and it now reads in line 249 to 263: "*While the single addition of Fe significantly enhanced growth of the bloom-forming species P. antarctica at West, the combined addition of FeMn did surprisingly not trigger its growth (Fig. 3c). Similarly, growth of P. antarctica was inhibited when Fe was added in combination with Zn and Co to the same phytoplankton community (Koch et al., unpublished data). Only when Fe was added together with vitamin B₁₂, maximum growth rate of P. antarctica was achieved at West, indicating relief from FeB₁₂ co-limitation (Koch et al., unpublished data). While we are unsure how to explain these effects, top-down and bottom-up controls such as growth inhibition and/or competition could be an explanation. The very low dFe concentrations at West (0.05 nM) potentially triggered competition for Fe among the different members of the community⁵⁰. As the bloom-forming diatoms are generally considered as r strategists²⁵, they usually have high resource uptake capabilities, thus providing them with a competitive advantage over other groups. Due to its much higher cell abundance relative to P. antarctica at West (Suppl. Table 2), the bloom-forming Fragilariopsis sp. may have been more efficient than the prymnesiophyte in taking up the added Fe and Mn, potentially causing the observed diminished growth rates by P. antarctica (Fig. 3c). Similar to our findings here, during a diatom-dominated bloom, strong competition for dFe between microbes, small phytoplankton and diatoms was reported⁵⁰.*"

R3_7: There is a brief sentence on this in the discussion, but I think that should be expanded a bit more. Is there some role for physiological acclimation, i.e. a change of chl-a per cell? How might that influence your results? → Please see response previously given to comment **R3_6**.

Line by line comments

R3_8: Line 102: specify that this is the final time point concentrations. → This was corrected in line 107: "... at the end of the experiment."

R3_9: Line 104: I don't know what these percentages in parentheses are referring to – I'm guessing it's 1.99 times the control? I think this could be a bit more precise. → Please see our response given to comment **R3_2**.

R3_10: Line 106: Specify that the +Fe was higher than the +FeMn. → The reviewer is right, this has been now clarified, it now reads in line 115: "*In comparison, the Chla-based accumulation rates of the +FeMn were lower than for the +Fe treatment.*"

R3_11: Line 109-111: This is misleading, since your statement should really be comparing +Fe to +FeMn, which from your data looks like there is no difference. → To account for the reviewer's comment, it now reads in line 114: "*For the small cells (< 2 μm), Chla-based accumulation rates of the Control were higher relative to the +Mn, lower relative to the +Fe and similar to the +FeMn treatment.*"

R3_12: Line 120: Refer to the actual changes in growth rather than percentages here, from looking at your figure it doesn't look like a huge change in Frag growth. → As suggested by the reviewer, we removed the magnitude of changes that was expressed in % from the revised manuscript. With respect to the growth response of *Fragilariopsis*, it is now written in line 122: "*The growth rate of Fragilariopsis sp. was similar between the Control and the +Mn treatment, but was significantly increased after addition of +Fe or +FeMn. In comparison, the addition of FeMn together resulted in a more significant increase in growth of Fragilariopsis sp. relative to the +Fe addition.*"

R3_13: Line 134-136: The effect size of the MnFe compared to Fe alone is very modest. → As suggested by the reviewer, we removed the magnitude of changes that was expressed in % from the revised manuscript.

R3_14: Paragraph 137: no reference to figure. → As suggested, we now refer to figure 4c in line 146.

R3_15: Line 141-142: refer to the actual growth rates, the effect size looks quite modest. → As suggested by the reviewer, we removed the magnitude of changes that was expressed in % from the revised manuscript.

R3_16: Line 155: The data from this paragraph should be presented as a figure, I think. For example, the difference between P in +Fe to +FeMn has a very small effect size, which would be evident from a figure. → We do agree that the photophysiological data in the table 3 do not allow well to recognize small changes. Significant changes are, however, indicated by different letters. Considering also that these changes were in most cases not of importance for the overall discussion of the data, we would like to keep showing these data in a table.

R3_17: Line 173: What does these lines represent? If you're going to fit a linear model, as these lines imply, that is a separate analysis than a correlation as you've described. → The lines represent a simple linear regression. To clarify this in the legend of figure 5, it now reads: "**Figure 5.** Linear regression (red line) between the dark-adapted maximum photosystem II quantum yield (F_v/F_m) and (a) the concentrations of total dissolved iron (dFe) or (b) the concentrations of total dissolved manganese (dMn).". In addition, information on R^2 , r and p is now also given in figure 5a and b. We first fitted our data with a simple linear regression to test the relationship of dTMs and F_v/F_m . In addition, we applied a Pearson correlation test to assess the intensity of the relationship between dTMs and F_v/F_m . This information is now provided in the line 499 in the **Methods** section, in line 184-189 of the **Results** section and as well as in the figure 5.

R3_18: Line 184: I do not agree that you have provided evidence Mn is a "major driver of Southern Ocean phytoplankton distribution". You didn't really look at phytoplankton distributions, so I'm not sure how you can claim that. Also, the statement that picoeukaryotes at station east and Frag at station west were "clearly colimited" I think is quite a stretch. Overall the additional benefit to inferred growth rate was quite patchy when comparing +Fe to +FeMn. → We agree that the word 'distribution' is wrong and thus it was removed in the whole manuscript. Please note that it was removed from line 192 and the whole part 2 of the discussion has been completely changed to discuss more thoroughly the different growth responses at both locations.

R3_19: Line 188: There needs to be a discussion about the disagreements between your two incubation experiments. → We totally agree with this suggestion. As a consequence, part 2 of the discussion (line 212 to 275) has been completely changed and we now discuss more thoroughly the different growth responses at both locations.

R3_20: Line 230: While I do agree, your language throughout on the effect of MnFe colimitation is too strong. → The wording was indeed too strong and has been changed throughout the revised manuscript.

R3_21: Line 251: This is a bit vague. What about downregulation of Chl-a per cell? To me that seems like the most obvious reason. → Please see our response given to comment **R3_6**.

R3_22: Line 289: 'based on Moore et al 2013'... is that really the takeaway? I could be misremembering, but the 1:1 ratios of dFe and dMn in Moore et al 2013 were evidence of which elements in the ocean tend to be limiting. In any case, it's not clear to me what a 1:1 ratio of dFe:dMn is optimizing? (Since you've used the word 'optimal'!) → Please note that this part has been removed from the discussion.

R3_23: Line 321: The language is way too strong for a correlation of 11 datapoints. → As suggested by the reviewer, we have weakened our statements in the revised manuscript (line 291-315). Moreover, we extended the dataset with 11 additional sampling stations, for which F_v/F_m as well as concentrations of dFe and dMn were also determined (Table 4) during another Polarstern expedition (PS112, 2018), and included them into the data analysis (linear regression and correlation as shown in **Figure 5**).

Figure 5. Linear regression (red line) between the dark-adapted maximum photosystem II quantum yield (F_v/F_m) and (a) the concentrations of total dissolved iron (dFe) or (b) the concentrations of total dissolved manganese (dMn). The data were determined at 22 in situ stations sampled across the Drake Passage and around the Western Antarctic Peninsula, including BIO1 as station West and BIO3 as station East. The two experimental stations West and East are indicated by dark circles while the PS97 stations are denoted by white circles and the PS112 stations by white triangles.

R3_24: Line 43: might be helpful to specify MnSOD? → As suggested, we have specified “*Mn-containing antioxidant enzyme superoxide dismutase*” in line 46.

R3_25: Line 360: GEOTRACES all caps → This was added in line 364.

R3_26: Line 386: “physiological health” is vague, maybe specify photophysiological or be more precise? → Indeed, the term “physiological health” was too vague as we monitored the photosynthetic efficiency of the cells. This was corrected in line 390: “*The photosynthetic efficiency ...*”

R3_27: Line 415: do these numbers represent standard error or standard deviation? → This numbers represent standard deviation and this information was added in line 420.

R3_28: Line 456: I believe there is a mistake in this equation? I don’t think there are two natural logarithms, only one. I think the equation should be: $\mu = (\ln \frac{N_2}{N_1}) / dt$ → The reviewer is right, there was a typo, which was corrected in line 451.

R3_29: Line 491: specify the response variable. → This is now specified in line 494: “... in order to assess the impact of Fe and Mn availability compared to the control treatment on cell abundance and Chla accumulation rate.”

R3_30: Line 499: equation 4 is mentioned but is not present anywhere in the text. → Indeed, equation 4 was missing. This have been added in line 503: “The Shannon diversity index (H) was calculated from: $H = -\sum_{i=1}^R p_i \ln p_i$ (Eq.3), where R is the total number of species and p_i the relative abundance of species.”

Reviewers' comments:

Reviewer #2 (Remarks to the Author):

The revised manuscript has been improved greatly. The authors have addressed most of my concerns and sufficiently responded to my comments.

I do not require to review the manuscript again but do however have a few last suggestions/corrections that the authors should consider:

- Supplementary figures SFig.2 and SFig.3 are added to the revised manuscript supplementary but are not referenced/used in the main text. I suggest adding a mention and reference to SFig.2 in figure captions of Figure 3 and Figure 4 and a reference to SFig.3 to line 160/161 and Table 3 caption.
- For newly added Figure 2, add a reference to STable 1 and STable2 (data these piechart figures are based on).

Reviewer #3 (Remarks to the Author):

Thank you to Balaguer and coauthors for their updated manuscript. Unfortunately, I still have some reservations about the interpretation of these data. I think that this manuscript would benefit from some additional proof reading, there are a few grammatical and sentence structure mistakes that make it a bit challenging to follow at times. The changes I am suggesting can be easily addressed, and I still think that this is a valuable contribution. Overall, I think that these data are overwhelmingly pointing to iron as the main driver of productivity, and only in a small number of cases Mn potentially influencing community structure. I voice these concerns only because I think that this is an important manuscript for the community of people thinking about TM limitation in the ocean!

Major comments:

Figures

I think these updated figures are much better, thank you!

Effect Size

I do not think my original comment about effect size was implemented. Effect size would mean reporting the actual effects – in mean growth rate values, etc. For example:

Line 124: Instead of saying “resulted in a significant increase”, my suggestion was to emphasize effect size. For example, “resulted in an increase of growth from XX to XX mean growth per treatment”.

Line 264: This is an example where effect size should be emphasized, which would help readers evaluate the experiments in Location East. “the analysis of the bulk parameter Chla indicated a Fe-Mn co-limitation effect at the community level”. From looking at your figure, the change is from control ~0.125 to 0.26 for +Fe, and 0.275 for +FeMn (i.e. an additional 0.0175 μ per day). By emphasizing effect size, the reader would understand that the additional effect (comparing +Fe to +FeMn) is modest.

Importance of Mn for Colimitation

My original comment was concerning the importance of Mn in the Southern Ocean. From your data, it looks like you found very small effects in terms of both bulk and community-resolution effects of +Mn

and +MnFe vs. +Fe. In some cases, certain community members were sensitive to a combination, but these effects were not consistent across your two locations. I do not feel that you have addressed one of my main concerns in the original review, where I wrote:

“Presumably if something is a major controller of SO phytoplankton, there should be some replicability when two experiments are conducted in the same region of the ocean? For example, to me the result that *Fragilariopsis* was Mn-Fe colimited in one experiment but not the other should indicate that Mn-Fe colimitation is actually not very common.”

Therefore I do not think your current manuscript title, nor the conclusions, are supported:

“Co-limitation between Fe and Mn may be more prominent in the SO than previously thought, with important implications for phytoplankton community structure. Therefore, dFe must not be the only TM considered when assessing future biogeochemical changes in this climate relevant region of the world ocean.”

Why would colimitation between Fe and Mn be more prominent than previously though? Previous work, even from Buma et al 1991, showed that the effect of Mn is patchy. Your data support that. Further, it only appears that sometimes there are any implications for phytoplankton community structure.

From the title, I do not agree that phytoplankton community composition is governed by manganese limitation. In almost every case for every variable of community composition, the control and +Mn were very similar. In a small number of cases that are not consistent across experiments, they were different.

In the revised discussion there is a large emphasis on these unpublished data from Koch et al. I cannot evaluate the conclusions that rely on these data. However, even the statement about addition of FeB12 is confusing – the growth of this combined addition (FeB12) was about 0.12 per day, which is almost identical to what is plotted in Figure 4C for the *Fragilariopsis* section +Fe? How would that indicate that B12 is also limiting then?

Overall I find the discussion from 236-243 to be speculative and difficult to evaluate. What about a simpler explanation: they have achieved maximal growth in the time that they had? Why do you need to invoke a complex scenario of colimitation by so many micronutrients?

Statistical analysis

Just to be clear in what I'm trying to communicate: a linear regression is an equation of the form:

$$Y = m*x + b$$

There are two parameters that are estimated, m and b. (The slope and the intercept). If you ran a linear regression, you should report the estimated slope and intercept, and the standard error around those coefficient estimates. Because you show a line in your figure of Fv/Fm with dMn, dFe, I assume you ran a linear regression, and should therefore report these values.

Pearson correlation compares the covariance two variables with their standard deviation. There is no parameter estimation, except you do end up with an r, R² and p value. The difference is that in a linear regression you are creating a model, i.e. in a Pearson correlation you cannot take this equation and predict some future values of Fv/Fm using the r, R² and p values. It is not clear in the manuscript which of these two (or both?) analyses you conducted.

Data availability

Thank you for providing the link to your data. I clicked on it and it says link not found, not sure if there is a mistake in this link or if it's just not available yet?

Minor comments:

Line 51: Martin et al 1990 did not show any colimitation between Mn and Fe (ref 7)

Line 88: not sure what "almost similar" means

Line 90: would be helpful to specify this is % total cells

Line 112: typo, or does "all" mean the same as "total" here?

Line 121: throughout, "sp." should not be italicized

Line 144: This is another opportunity to specify the increase in growth from +Mn to +MnFe.

Line 150: Typo

Line 179: Typo

Line 540: I think it would be helpful if this table included depths, I'm not sure if they are reported in the manuscript anywhere?

Line 227: Typo

Line 309: "might not only be a proxy for Fe limitation". Doesn't your data say the opposite, that it probably is not a good proxy for Fe limitation?

Dear reviewers,

we have responded to all comments and have revised the manuscript that was entitled “*Phytoplankton species composition is governed by both iron and manganese limitation in the Drake Passage*” again. We would like to thank you for the helpful comments, which greatly increased the quality of the paper. Our responses are below and marked in **purple**. We have highlighted all changes in **yellow** colour in the denoted file “Article”. Please note that now the manuscript title has been changed to: “*Iron and manganese co-limit the growth of two phytoplankton groups dominant at two locations of the Drake Passage*” in accordance to the comments below.

Response to reviewers' comments:

Reviewer #2 (Remarks to the Author):

The revised manuscript has been improved greatly. The authors have addressed most of my concerns and sufficiently responded to my comments.

→ We thank reviewer 2 for the helpful comments and we are pleased that we could address most of the concerns.

I do not require to review the manuscript again but do however have a few last suggestions/corrections that the authors should consider:

- Supplementary figures SFig.2 and SFig.3 are added to the revised manuscript supplementary but are not referenced/used in the main text. I suggest adding a mention and reference to SFig.2 in figure captions of Figure 3 and Figure 4 and a reference to SFig.3 to line 160/161 and Table 3 caption. → As suggested, Suppl. Fig.3 is now indicated in line 155 and is also mentioned line 389 in the Methods section. In addition, Suppl. Fig.2 is now also mentioned in the captions of figure 3 and 4.

For newly added Figure 2, add a reference to STable 1 and STable2 (data these piechart figures are based on). → As suggested, references to STable 1 and 2 were added in the caption of Figure 2.

Reviewer #3 (Remarks to the Author):

Thank you to Balaguer and coauthors for their updated manuscript. Unfortunately, I still have some reservations about the interpretation of these data. I think that this manuscript would benefit from some additional proof reading, there are a few grammatical and sentence structure mistakes that make it a bit challenging to follow at times. The changes I am suggesting can be easily addressed, and I still think that this is a valuable contribution. Overall, I think that these data are overwhelmingly pointing to iron as the main driver of productivity, and only in a small number of cases Mn potentially influencing community structure. I voice these concerns only because I think that this is an important manuscript for the community of people thinking about TM limitation in the ocean!

Major comments:

Figures

I think these updated figures are much better, thank you!

Effect Size

I do not think my original comment about effect size was implemented. Effect size would mean reporting the actual effects – in mean growth rate values, etc. For example: Line 124: Instead of saying “resulted in a significant increase”, my suggestion was to emphasize effect size. For example, “resulted in an increase of growth from XX to XX mean growth per treatment”. → We agree that even though we removed the magnitude of changes that was expressed in % from the revised manuscript, the effect size was not yet implemented. To account for this, we have modified the respective sections as suggested by the reviewer and it now reads in:

- Line 122-123: *“In comparison, the addition of FeMn together resulted in a significant increase of growth of *Fragilariopsis* sp. from 0.08 to 0.11 day⁻¹ relative to the +Fe addition.”*
- Line 135-136: *“When Fe and Mn were added together, the Chla-based accumulation rate of all and small cells significantly increased from 0.02 to 0.14 day⁻¹ relative to the addition of Fe alone.”*
- Line 140-141: *“The addition of TMs together led to a significant growth increase of P2 from 0.11 to 0.13 day⁻¹ compared to the single addition of Fe.”*

Line 264: This is an example where effect size should be emphasized, which would help readers evaluate the experiments in Location East. “the analysis of the bulk parameter Chla indicated a Fe-Mn co-limitation effect at the community level”. From looking at your figure, the change is from control ~0.125 to 0.26 for +Fe, and 0.275 for +FeMn (i.e. an additional 0.0175 mu per day). By emphasizing effect size, the reader would understand that the additional effect (comparing +Fe to +FeMn) is modest. → As suggested by the reviewer, we have now emphasized the effect size in line 262-267: *“While the effect of FeMn compared to Fe alone had a limited effect on total Chla (higher by 0.02 d⁻¹), the small cells (< 2 μm) reached highest values only when both Fe and Mn were added (higher by 0.14 d⁻¹, Fig.4a). A modest, but significant growth stimulation by 0.02 day⁻¹ after addition of Fe and Mn together relative to Fe alone was also observed for the picoeukaryote group P2, which numerically dominated the East community (Fig.4b) and thus contributed most to the Chla pool (Fig.4a).”*.

We also modified the discussion part related on Chla-specific growth rates of the community at location West, it now reads in line 221-225: *“Even though for experiment West, the analysis of bulk parameters such as Chla indicated Fe availability as main driver, the microscopic analysis, however, revealed that the ecologically and biogeochemically important diatom *Fragilariopsis* sp., which dominated the microphytoplankton community, significantly increased its growth by 0.03 day⁻¹ only when Fe was added in conjunction with Mn compared to the single addition of Fe (Fig. 3c).”*.

Importance of Mn for Colimitation

My original comment was concerning the importance of Mn in the Southern Ocean. From your data, it looks like you found very small effects in terms of both bulk and community-resolution effects of +Mn and +MnFe vs. +Fe. In some cases, certain community members were sensitive to a combination, but these effects were not consistent across your two locations. I do not feel that you have addressed one of my main concerns in the original review, where I wrote: “Presumably if something is a major controller of SO phytoplankton, there should be some replicability when two experiments are conducted in the same region of the ocean? For example, to me the result that *Fragilariopsis* was Mn-Fe colimited in one experiment but not the other should indicate that Mn-Fe colimitation is actually not very common.”

→ With respect to the reviewer's comment on the replicability of the outcome of both experiments, we still believe that at each location a key phytoplankton group was identified as being FeMn co-limited. Please note that the identified Fe-Mn phytoplankton group at each location numerically dominated, hence they represented key members of the investigated phytoplankton communities. The latter finding is of importance, as it indicates that important members with high cell abundances can indeed be Fe-Mn limited even though other members of the community are primarily limited by Fe alone. The latter is indeed a new finding that has never been observed before and points towards complex species interaction effects that drive phytoplankton community composition at specific locations in the SO, as in our study in the Drake Passage. To account for this point, it now reads: At location West, it was "*Fragilariopsis* sp., which numerically dominated the microphytoplankton community" (line 223-224) and "*the picoeukaryote group P2, which numerically dominated the East community*" (line 265-266). To make this point clearer in the revised manuscript, this has been modified throughout the manuscript:

- Line 20-21: "... of a key phytoplankton group at each location: at West the dominant diatom *Fragilariopsis* and one subgroup of picoeukaryotes, which numerically dominated the East community."

- Line 77-78: "... of two key members of the phytoplankton community (*Fragilariopsis* sp. and picoeukaryotes)."

- Line 186-187: "... *Fragilariopsis* sp., being the most abundant species among the microphytoplankton community at the location West, as well as picoeukaryotes, which dominated the whole community at the location East ..."

- Line 221-227: "Even though for experiment West, the analysis of bulk parameters such as *Chla* indicated Fe availability as main driver, the microscopic analysis, however, revealed that the ecologically and biogeochemically important diatom *Fragilariopsis* sp., which dominated the microphytoplankton community, significantly increased its growth by 0.03 day⁻¹ only when Fe was added in conjunction with Mn compared to the single addition of Fe (Fig. 3c). In fact, the cell abundance of *Fragilariopsis* sp. was enhanced by ~35% in the +FeMn relative to the +Fe treatment (Suppl. Table 2). This suggests a relief of FeMn co-limitation of the dominant diatom *Fragilariopsis* sp. at West."

- Line 264-267: "A modest, but significant growth stimulation by 0.02 day⁻¹ after addition of Fe and Mn together relative to Fe alone was also observed for the picoeukaryote group P2, which numerically dominated the East community (Fig.4b) and thus contributed most to the *Chla* pool (Fig.4a)."

The reviewer is right that the explanation why *Fragilariopsis* was Mn-Fe co-limited at location West, but not at location East, was still not provided. To account for this, part 2 of the discussion has been changed from line 240 to 245 was added:

"Due to the reason that *F. curta*, *F. nana*, *F. cylindrus* and *F. pseudonana* are very difficult to distinguish under the light microscope⁴⁹, we grouped these species together as *Fragilariopsis* sp.. Therefore, the ratio between them at both locations is unknown. We speculate that the TMs requirements by the same or different *Fragilariopsis* species may potentially differ, as previously observed for different *Pseudo-nitzschia* species⁵⁰. Unfortunately, almost nothing is known on the inter- and intraspecific TM requirement of different *Fragilariopsis* species." We hope this explanation clarifies the different responses observed for *Fragilariopsis* at both locations.

New reference:

⁴⁹Cefarelli, A. O. *et al.* Diversity of the diatom genus *Fragilariopsis* in the Argentine Sea and Antarctic waters: morphology, distribution and abundance. *Polar biology* **33**, 1463–1484 (2010).

⁵⁰Marchetti, A. & Harrison, P. J. Coupled changes in the cell morphology and elemental (C, N, and Si) composition of the pennate diatom *Pseudo-nitzschia* due to iron deficiency. *Limnology and Oceanography* **52**, 2270–2284 (2007).

Therefore, I do not think your current manuscript title, nor the conclusions, are supported: “Co-limitation between Fe and Mn may be more prominent in the SO than previously thought, with important implications for phytoplankton community structure. Therefore, dFe must not be the only TM considered when assessing future biogeochemical changes in this climate relevant region of the world ocean.”

Why would colimitation between Fe and Mn be more prominent than previously though? Previous work, even from Buma *et al.* 1991, showed that the effect of Mn is patchy. Your data support that. Further, it only appears that sometimes there are any implications for phytoplankton community structure.

→ The reviewer is right that our title is incorrect. Indeed, it is not the effect of Mn alone as previously suggested by our title in the last revised manuscript. In line with Buma *et al.* (1991), the single addition of Mn in our study has a negligible effect on the species composition. As the previous title was indeed misleading, to specify that Mn only in combination with Fe has an effect, we have now changed it to: “*Iron and manganese co-limit the growth of two phytoplankton groups dominant at two locations of the Drake Passage*”.

According to the reviewer that the statement “*Co-limitation between Fe and Mn may be more prominent in the SO than previously thought, with important implications for phytoplankton community structure.*” is not supported by our data, we agree that this sentence was misleading. Hence, it was removed in line 321 and to clarify what we wanted to say, it now reads in line 329-334: “*As only few FeMn-enrichment experiments were performed in SO waters so far^{22,23}, the occurrence of FeMn co-limitation of phytoplankton communities across the SO has not yet thoroughly been assessed. Therefore, one could speculate that within an Fe-limited phytoplankton community some phytoplankton species or groups could be still Fe-Mn co-limited, remaining, however, yet undetected due to a lack of experiments and thus little data availability.*”

Moreover, we removed the sentence in line 344-345 “*Co-limitation between Fe and Mn may be more prominent in the SO than previously thought, with important implications for phytoplankton community structure.*” We, however, still keep the sentence that “*dFe must not be the only TM considered*” due to the reason express in the answer above.

From the title, I do not agree that phytoplankton community composition is governed by manganese limitation. In almost every case for every variable of community composition, the control and +Mn were very similar. In a small number of cases that are not consistent across experiments, they were different. → We agree and, as mentioned above, we have rephrased the title to: “*Iron and manganese co-limit the growth of two phytoplankton groups dominant at two locations of the Drake Passage*”.

In the revised discussion there is a large emphasis on these unpublished data from Koch *et al.* I cannot evaluate the conclusions that rely on these data. However, even the statement about addition of FeB12 is confusing – the growth of this combined addition (FeB12) was about 0.12

per day, which is almost identical to what is plotted in Figure 4C for the *Fragilariopsis* section +Fe? How would that indicate that B12 is also limiting then?

Overall I find the discussion from 236-243 to be speculative and difficult to evaluate. What about a simpler explanation: they have achieved maximal growth in the time that they had? Why do you need to invoke a complex scenario of colimitation by so many micronutrients?

→ We agree with the reviewers that these additional data bring confusion, therefore we removed this dataset in the revised manuscript (from line 231 to 237 and 247 to 251) and adapted our argumentation from line as mentioned above.

Statistical analysis

Just to be clear in what I'm trying to communicate: a linear regression is an equation of the form:

$$Y = m \cdot x + b$$

There are two parameters that are estimated, m and b . (The slope and the intercept). If you ran a linear regression, you should report the estimated slope and intercept, and the standard error around those coefficient estimates. Because you show a line in your figure of F_v/F_m with dMn , dFe , I assume you ran a linear regression, and should therefore report these values. Pearson correlation compares the covariance two variables with their standard deviation. There is no parameter estimation, except you do end up with an r , R^2 and p value. The difference is that in a linear regression you are creating a model, i.e. in a Pearson correlation you cannot take this equation and predict some future values of F_v/F_m using the r , R^2 and p values. It is not clear in the manuscript which of these two (or both?) analyses you conducted. → We agree with the reviewer that it was not clear which analyses were done in the previous version of the manuscript and in addition the linear equation was not provided. We have performed both analyses as stated in line 483 in the Methods section. Please also note that the red lines in figure 5 are linear regressions, this information is also specified in the figure caption. In figure 5, we now show in addition to the R^2 (=Variance explained by the model/Total variance), the estimated slope and intercept extracted from the linear regression.

The independent results of the Pearson correlation tests are described in the results section in line 177: "...therefore there was no correlation between dFe and F_v/F_m (Fig. 5a ; $r \sim 0.2$, $p > 0.05$)." The figure 5 have been now updated. We hope this clarifies, which analyses have been done.

Data availability

Thank you for providing the link to your data. I clicked on it and it says link not found, not sure if there is a mistake in this link or if it's just not available yet? → The data have been approved in PANGAEA and therefore the link provided in the manuscript will work as soon as the paper will be published. Currently, the data are not yet accessible, the reviewer is right, as they are still under an embargo until acceptance of this manuscript. We do not want the data to be available on PANGAEA before the paper is published.

Minor comments:

Line 51: Martin et al 1990 did not show any colimitation between Mn and Fe (ref 7) → We agree with the reviewer, this reference was removed.

Line 88: not sure what “almost similar” means → We agree that the term “almost similar” is extremely unclear. Please note that this sentence has been removed.

Line 90: would be helpful to specify this is % total cells → This was specified in line 90.

Line 112: typo, or does “all” mean the same as “total” here? → Indeed, “all” meant total but this was now changed in line 111: “For all (total) and...”.

Line 121: throughout, “sp.” should not be italicized → This has been corrected throughout the manuscript.

Line 144: This is another opportunity to specify the increase in growth from +Mn to +MnFe. → This has been changed to in lines 140-141: “The addition of TMs together led to a growth increase of P2 from 0.11 to 0.13 day⁻¹ compared to the single addition of Fe.”.

Line 150: Typo → This has been corrected through all the manuscript.

Line 179: Typo → This has been corrected in line 171.

Line 540: I think it would be helpful if this table included depths, I'm not sure if they are reported in the manuscript anywhere? → Please note that the sampling depth at each sampling location was mentioned in the Methods section in line 354 and 356.

Line 227: Typo → *Chla* is all along written with an 'a' in italic.

Line 309: "might not only be a proxy for Fe limitation". Doesn't your data say the opposite, that it probably is not a good proxy for Fe limitation? → No, it was not our intention to state that F_v/F_m is not a good indicator for Fe limitation. To avoid this impression, this sentence has been modified to (309-310): "*... F_v/F_m may also be a photophysiological signature for a TM co-limitation with Fe, in our case for Fe and Mn.*"